# The negative cofactor 2 complex is a key regulator of drug resistance in *Aspergillus fumigatus*

Takanori Furukawa [1,2,7], Norman van Rhijn [1,2,7], Marcin Fraczek[1], Fabio Gsaller[1], Emma Davies[1], Paul Carr[1], Sara Gago [1,2], Rachael Fortune-Grant[1,2], Sayema Rahman[1,2], Jane Mabey Gilsenan[1], Emma Houlder [2], Caitlin H. Kowalski[3], Shriya Raj [4], Sanjoy Paul[5], Peter Cook [2], Josie E. Parker [6], Steve Kelly [6], Robert A. Cramer[3], Jean-Paul Latgé[4], Scott Moye-Rowley[5], Elaine Bignell[1,2], Paul Bowyer[1,2,8]* & Michael J. Bromley [1,2,8]*

The frequency of antifungal resistance, particularly to the azole class of ergosterol biosynthetic inhibitors, is a growing global health problem. Survival rates for those infected with resistant isolates are exceptionally low. Beyond modification of the drug target, our understanding of the molecular basis of azole resistance in the fungal pathogen *Aspergillus fumigatus* is limited. We reasoned that clinically relevant antifungal resistance could derive from transcriptional rewiring, promoting drug resistance without concomitant reductions in pathogenicity. Here we report a genome-wide annotation of transcriptional regulators in *A. fumigatus* and construction of a library of 484 transcription factor null mutants. We identify 12 regulators that have a demonstrable role in itraconazole susceptibility and show that loss of the negative cofactor 2 complex leads to resistance, not only to the azoles but also the salvage therapeutics amphotericin B and terbinafine without significantly affecting pathogenicity.

[1] Manchester Fungal Infection Group, Division of Infection, Immunity and Respiratory Medicine, Faculty of Biology, Medicine and Health, University of Manchester, CTF Building, 46 Grafton Street, Manchester M13 9NT, UK. [2] Lydia Becker Institute of Immunology and Inflammation, Manchester Collaborative Centre for Inflammation Research, Division of Infection, Immunity and Respiratory Medicine, Faculty of Biology, Medicine and Health, University of Manchester, Manchester Academic Health Science Centre, Manchester, UK. [3] Department of Microbiology and Immunology, Geisel School of Medicine at Dartmouth, Hanover, NH 03766, USA. [4] Unité des Aspergillus, Institut Pasteur, 25 rue du Docteur Roux, 75724 Cedex 15 Paris, France. [5] Department of Molecular Physiology and Biophysics, Carver College of Medicine, University of Iowa, Iowa City, IA 52242, USA. [6] Institute of Life Science, Swansea University Medical School, Swansea University, Swansea, Wales SA2 8PP, UK. [7] These authors contributed equally: Takanori Furukawa, Norman van Rhijn. [8] These authors jointly supervised this work: Paul Bowyer, Michael J. Bromley. *email: Paul.Bowyer@manchester.ac.uk; Mike.Bromley@manchester.ac.uk

*A*spergillus fumigatus is the most important airborne mould pathogen and allergen worldwide. Estimates suggest that over 3 million people have invasive or chronic infections that lead to in excess of 600,000 deaths every year[1]. Only three classes of drugs are currently recommended for the treatment of aspergillosis with the azole class being recommended for primary therapeutic purposes, and amphotericin B and the echinocandins (caspofungin and micafungin) for salvage therapy. With effective treatment, mortality rates for invasive disease remain ~50%[2]. It is of great concern, however, that drug resistance to the azoles is rapidly emerging. For individuals that are infected with a resistant isolate the mortality rate exceeds 80%[3,4]. As disease progression is so rapid in invasive aspergillosis (IA)[5], therapy failure is attributable to delays in administering alternative agents. Methods to rapidly detect resistance are critical to ensure effective transition to alternative appropriate therapies.

Our understanding of the factors governing azole resistance in *A. fumigatus* is not fully defined. The azoles act by inhibiting the hemoprotein lanosterol demethylase (Cyp51A) resulting in a reduction of the key sterol, ergosterol, in the fungal membrane and an accumulation of toxic sterol intermediates[6]. Azole resistance is frequently associated with an allelic variant of *cyp51A* that incorporates a tandem repeat in the promoter, typically $TR_{34}$ or $TR_{46}$, with a secondary non-synonymous mutation within its coding sequence[7]. These modifications appear to have no substantial impact on pathogenicity in murine models of invasive aspergillosis[8]. The mechanism of resistance in a significant proportion of other isolates remains unclear[7]. This hinders the development of rapid molecular diagnostics to detect drug resistance in the clinic. Our limited understanding of the mechanisms of azole resistance also prevents the development of combination therapeutic strategies that specifically target drug-resistance mechanisms[9].

We and others have recently reported on the role played by various *A. fumigatus* transcriptional regulators in response to azole antifungal drugs[10–12]. The sterol regulatory element-binding protein (SREBP), SrbA is a basic helix–loop–helix (bHLH) transcriptional activator which directly regulates at least seven genes in the ergosterol biosynthetic pathway, including *cyp51A*[10]. Loss of *srbA* through gene replacement results in a significant increase in susceptibility to azoles[13] in part due to significant reductions in *cyp51A* mRNA levels[14]. The binding site for SrbA in the *cyp51A* promoter falls within the 34 and 46 mers duplicated in TR34/46 pandemic azole-resistant isolates[12]. The repeat duplicates the DNA-binding site leading to SrbA-mediated upregulation of *cyp51A* and a concomitant increase in azole resistance. AtrR, a Zn2-Cys6 transcription factor, also positively regulates sterol biosynthesis and directly binds the *cyp51A* promoter at the TR site, and additionally the promoter of an azole exporter, *cdr1B*[11,15]. The CCAAT-binding domain complex CBC, a heterotrimer comprising HapB, HapC and HapE, is a negative regulator of sterol biosynthesis directly binding the promoters of 14 ergosterol biosynthetic genes, including *cyp51A*[12]. Loss of CBC function leads to pan-azole resistance. Notably, a clinical azole-resistant isolate with a defect in HapE that results in perturbed DNA binding at the *cyp51A* promoter has been described[12,16]. Binding of the CBC at the *cyp51A* promoter is facilitated by another transcriptional regulator, HapX[12]. HapX is an iron-responsive basic region leucine zipper (bZIP) transcription factor that regulates the expression of genes linked to iron acquisition, storage and metabolism and facilitates binding of the CBC[17]. It is also notable that loss of SrbA, the CBC, HapX or AtrR is associated with significant reductions in virulence in murine models of invasive pulmonary aspergillosis[11–13,17].

The transcriptional network governing azole resistance is therefore highly complex and involves multiple regulators, some of which remain to be identified as our current models of this network are not able to explain all of the existing clinically significant mechanisms of azole resistance. We therefore postulated that other perturbations of the transcriptional network would lead to alterations in azole resistance without affecting pathogenicity.

In this study, we have generated and screened a library of 484 *A. fumigatus* transcription factor null mutant strains and identified a cohort of 12 factors that govern azole resistance and sensitivity. Here, we describe in detail the role of two CBF/NF-Y family transcription regulators, AFUB_029870 (NctA) and AFUB_045980 (NctB) where loss of function leads to azole resistance. We show that the *A. fumigatus* NctA and NctB (Negative cofactor two A and B) are part of the same transcriptional regulatory complex and demonstrate that the NCT complex is a key regulator of ergosterol biosynthesis and the azole exporter CDR1B. We also report that loss of the NCT complex leads to a multi-drug-resistance phenotype, including the azoles (itraconazole, voriconazole and posaconazole) as well as the salvage therapeutic amphotericin B[18] and terbinafine, an agent used in the treatment of chronic and allergic disease[19]. Furthermore, loss of this complex results in a notable increase in the immunogenic properties of *A. fumigatus*, but does not result in loss of virulence. The results of our study highlight that loss of function of a single gene can give rise to cross-resistance to many of the primary drugs used to treat aspergillosis and should prompt clinical centres to look for these mechanisms of resistance. As we also show that strains lacking a functional NCT complex are hypersensitive to the echinocandins, switching patients harbouring strains with these mutations to caspofungin could be a preferred course of action.

## Results

**Generation of a library of transcription factor null mutants**. A systematic review of the genes previously annotated as transcription factors (TFs) in the databases at ENSEMBL fungi, ASPGD and DBD[20] resulted in the identification of 495 putative TFs (Supplementary Data 1). To further characterise this cohort, we classified each TF according to its Pfam domains, identified using Hidden Markov Model profiling (hmmscan). The majority of TFs were shown to have either one ($n = 245$) or two ($n = 158$) functional domains associated with transcriptional regulation (Supplementary Fig. 1). The most common domains identified were binuclear zinc cluster ($n = 192$ (PFAM00172) and fungal-specific transcription domains ($n = 195$ (PFAM04052 and 11951); Supplementary Fig. 1). This is consistent with previous reports that describe domains of transcription factors in ascomycetes[21]. Sixty-two proteins lacked any Pfam domains associated with transcription factor function. However, following cross-referencing of these to orthologues in the AF293 genome (whose genome annotation has been enhanced by comparison to RNA-seq data) to reconcile annotation differences, and additional evaluation to identify PROSITE, SUPERFAMILY, SMART and CDD domains, we were able to identify transcription factor-associated domains for all, but six proteins (Supplementary Data. 1). For two of these proteins, MedA and LaeA, a functional role in transcriptional regulation has been defined experimentally[22–24]. We were unable to identify any functional domains in the orthologue of the TFIIIC subunit *tfc6* or three proteins annotated as RfeD, F and G.

Unlike in the model yeast, *Saccharomyces cerevisiae*, gene replacement strategies in *A. fumigatus* are complicated by relatively low levels of homologous recombination. This problem can be mitigated by the use of strains lacking components of the non-homologous end joining machinery[25], such as Ku70[26],

Ku80[27] and Lig4[28], however, even in strains lacking these factors, gene replacement cassettes require around 1 kb of homologous sequence flanking each side of a target gene[27]. To facilitate the disruption of all of the TFs identified, we chose to employ a fusion PCR approach similar to that described by Szewczyk et al.[29] (see schematic Supplementary Fig. 2) and used custom developed scripts to design the primers for amplification of the gene replacement cassettes (see the Methods section). Cassettes were successfully amplified for all 495 of the TFs, and were used to transform MFIG001, a $\Delta ku80$, pyrG+ strain derived from FGSC strain A1160[30]. We isolated null homokaryons for 97.7% (484) transcription factor genes as defined by our ability to isolate strains, in which we could amplify from the hygromycin resistance cassette to a region beyond the gene replacement cassette in addition to a lack of a PCR product corresponding to the target gene (see schematic Supplementary Fig. 2). Precise replacement was further confirmed for a randomly selected subset of 12 mutants by Southern blot analysis (Supplementary Data 1). Despite several attempts (minimum $n = 3$), we were unable to isolate null mutants for 11 genes (Supplementary Data 1). The majority of these genes encode components of the RNA polymerase I/II/III transcription factor complexes, or are transcription factors, which locate at a higher-level of a regulatory hierarchy[31] whose deletion would likely to cause a lethal phenotype.

**An azole susceptibility screen of the transcription factor mutant library**. The minimum inhibitory concentrations (MIC) of itraconazole for the isogenic strain (MFIG001) was determined to be 0.5 mg/L in the RPMI-1640 media, following procedures outlined by EUCAST[32]. To identify novel transcriptional regulators associated with azole sensitivity and resistance, we screened the transcription factor null library at itraconazole concentrations representing sub-minimum inhibitory concentrations (MIC) (0.06 and 0.12 mg/L) and at the MIC (0.5 mg/L).

Six transcription factor null mutants exhibited clear and reproducible fitness defects in sub-MIC levels of itraconazole when compared with the cohort of mutants in the collection (Fig. 1a). The transcription factors knocked out in two of these mutants have previously been defined as activators of the ergosterol biosynthetic pathway, including cyp51A, namely SrbA and AtrR. The other null mutants identified in the screen lacked the carbon catabolite repressor CreA[33], the calcium-responsive regulator ZipD[34], the SAGA complex subunit AdaB[35] and the orthologue of the S. cerevisiae stress-responsive regulator GIS2 herein described as GisB[36]. When the library was screened at itraconazole levels at the MIC, we identified six transcription factor mutants, two of which we have previously described ($\Delta hapX$ and $\Delta hapB$)[12] and four of which we associate with azole resistance in A. fumigatus for the first time ($\Delta nctA$, $\Delta nctB$, $\Delta areA$ and $\Delta rscE$).

We extended our phenotypic profiling of these isolates to assess their general growth fitness (Supplementary Fig. 3) and sensitivity profiles to additional antifungal drugs (Fig. 1b, c). The identified transcription factor null mutants showed variable growth fitness profiles, depending on the culture conditions tested. However, we did not see a clear correlation between their growth fitness and azole-sensitivity phenotypes. In comparison with the isogenic wild-type control, $\Delta creA$ and $\Delta zipD$ showed increased sensitivity to the ergosterol biosynthetic inhibitor terbinafine. Interestingly, the nctA and nctB null mutants phenocopied each other (Supplementary Fig. 3) and were resistant to the triazoles voriconazole (>32-fold increase in MIC) and posaconazole (>128-fold increase), terbinafine (twofold increase in MIC),

miltefosine (eightfold increase in MIC) and amphotericin B (eightfold increase in MIC). Conversely, the nctA and nctB null mutants showed hypersensitivity to the cell wall perturbing agents Congo Red, Calcofluor White, caspofungin and micafungin.

**NctA and NctB are members of the CBF/NF-Y family of regulators**. Pfam domain searches indicate that NctA and NctB are members of the evolutionarily conserved CBF/NF-Y family of transcription factors, which include the CBC transcription regulator complex (Supplementary Fig. 4)[37]. NctA (AFUB_029870) encodes a 247 aa protein, and is a reciprocal BLAST match of the S. cerevisiae negative cofactor 2 (NC2) complex α-subunit Bur6. NctA and Bur6, however, show little sequence similarity with the exception of the CBF/NFYB domain where they share 49% identity over 75 contiguous amino acids. NctA has two paralogues in A. fumigatus, HapE and an as yet un-named regulator encoded by AFUB_058240. Consistent with the similar phenotypes of the nctA and nctB null mutants, nctB (AFUB_045980) encodes a 142 aa protein that is the reciprocal BLAST match of the S. cerevisiae NC2 complex β-subunit (Ncb2) sharing 49% sequence identity over 86% of the protein. NctB has one readily identifiable paralogue in A. fumigatus, HapC.

**Loss of NctA leads to a reduction in growth rate**. To determine if the effects of nctA loss were due solely to the gene replacement and not other mutagenic events from transformation, an nctA reconstituted strain was generated. The growth rate of the nctA null alongside the parent strain MFIG001 (WT) and the nctA reconstituted strain (nctA rec) was assessed on Aspergillus complete media (ACM), Aspergillus minimal media (AMM), RPMI-1640 and DMEM. In all conditions, the nctA null mutant had a significantly reduced growth rate (36–44% reduction) compared with MFIG001 (Fig. 2a, b). Importantly, the reconstituted strain was indistinguishable in these assays from MFIG001. This reduction in growth rate is associated with a decrease in the initial rate of germination (5–6 h) and hyphal extension (Fig. 2c, d).

**NctA and NctB co-operatively regulate the same network of genes**. To assess the role of NctA and NctB on global regulation of gene expression in A. fumigatus, we carried out transcriptomic analysis (RNA-seq) using RNA extracted from cultures grown in the RPMI-1640 media. In the absence of itraconazole, 1244 genes were upregulated (>2-fold; FDR < 0.05) and 1049 (>2-fold; FDR < 0.05) genes were downregulated in the nctA null when compared with the isogenic parent strain (Supplementary Data 2). Similarly, 1183 genes were upregulated and 1104 genes were downregulated in the nctB null mutant. In the presence of itraconazole, 735 genes were upregulated and 736 genes were downregulated in the nctA null mutant, and 865 genes were upregulated and 852 genes were downregulated in the nctB null mutant. Direct comparison of the two data sets shows that the regulons of both transcription factors have a very high degree of similarity in both conditions (Fig. 3a, b), suggesting that both transcription factors work co-operatively.

To assess if NctA physically interacts with NctB, and to determine if any other proteins are found complexed with NctA, we generated a strain which encoded a functional C-terminally S-tagged derivative of NctA (as shown by the fact that the strain that was indistinguishable from the wild-type isolate) (NctA-S-tag, see Supplementary Fig. 5), and performed co-immunoprecipitation followed by LC/MS identification of interacting proteins. In addition to NctA, 20 unique proteins were identified (Supplementary Table 1; >2 matched peptides), including NctB, the TBP-associated transcriptional regulator Mot1, the transcriptional

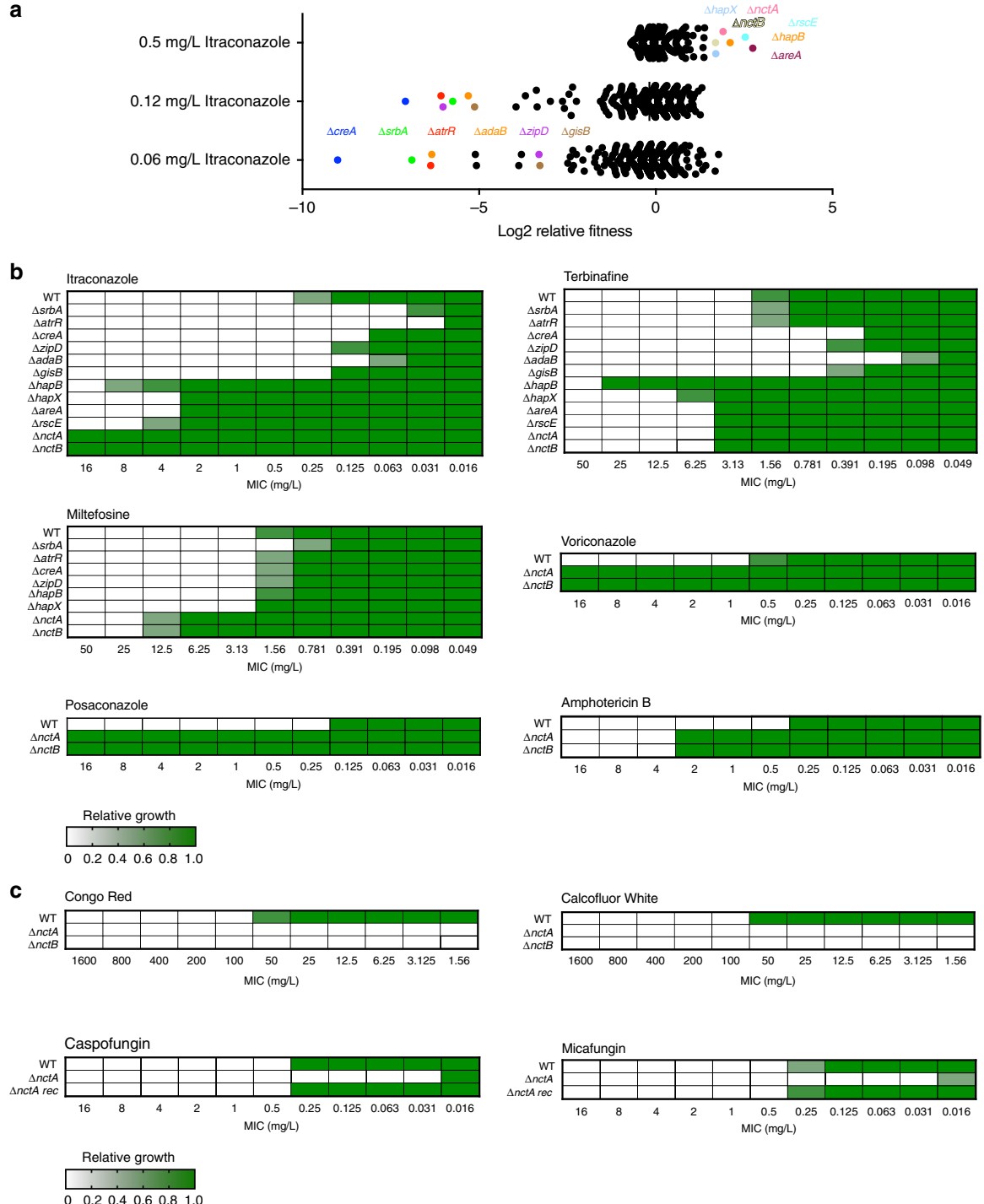

**Fig. 1 Identification of transcription factors associated with azole tolerance in *A. fumigatus*. a** Relative growth fitness of the 484 constructed TFKO mutants in MIC (0.5 mg/L) and sub-MIC (0.06 and 0.12 mg/L) levels of itraconazole. Each dot represents the mean of three replicate experiments. The TFKOs which exhibited significant fitness defects (fitness less than −4 for sub-MIC, and more than 1.5 for MIC condition) are indicated by coloured circles. **b, c** Heatmaps showing drug susceptibility profiles of the screened TFKO mutants to different antifungal drugs. Susceptibility assays were performed in triplicate, where optical density readings of fungal growth were standardised to no-drug control wells and shown as a relative growth values. Source data are provided as a Source Data file.

co-repressor Cyc8 and 9 ribosomal or ribosome-associated proteins. The interaction between NctA and NctB and Mot1 was supported with reciprocal co-immunoprecipitation using a C-terminally S-tagged version of NctB (NctB-S-tag, see Supplementary Fig. 5 and Supplementary Table 1). Taken together these data suggest that, consistent with the role of Bur6 and Ncb2 in

yeast[38,39], NctA and NctB form a complex with the TBP-associated co-regulator Mot1 in *A. fumigatus* and are responsible for regulating the same cohort of genes.

**The NCT complex is a global regulator of sterol biosynthesis.** The negative cofactor complex has been defined as a general

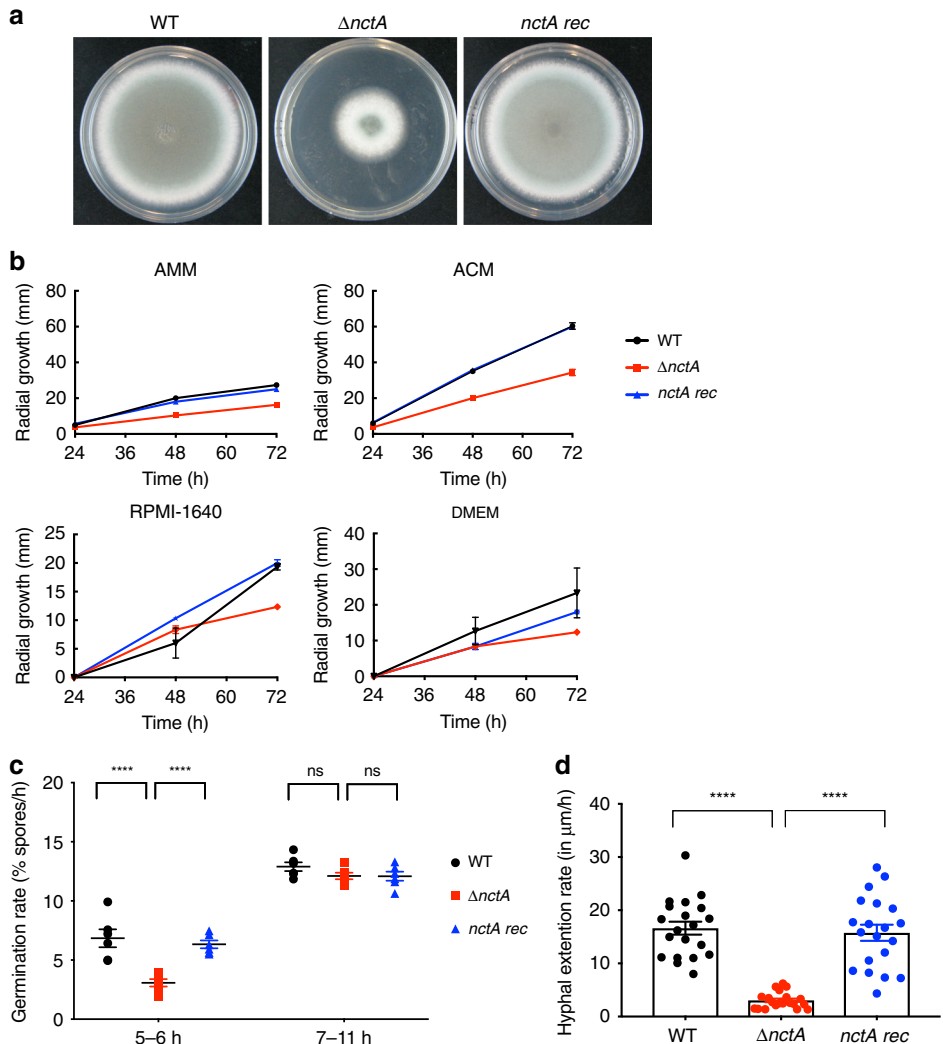

**Fig. 2 Impact of *nctA* deletion upon *A. fumigatus* growth. a** Colonial growth phenotypes of the wild-type MFIG001 (WT), the *nctA* null (Δ*nctA*) and the *nctA* reconstituted strain (*nctA rec*) grown on a solid Aspergillus complete medium (ACM) for 72 h at 37 °C. **b** Radial growth phenotypes on solid ACM, Aspergillus minimal medium (AMM), RPMI-1640 and DMEM. The error bars represent standard error of the mean of three independent experiments. **c** Germination rates in liquid RPMI-1640 at 37 °C. *P*-value was calculated by repeated measures two-way ANOVA with Sidaks correction: ****$P < 0.0001$; NS, $P > 0.05$. The error bars signify the standard deviations. **d** Hyphal extension rates in RPMI-1640 at 37 °C. *P*-value was calculated by Kruskal–Wallis test with Dunn's correction: ****$P < 0.0001$; NS, $P > 0.05$. Percentages of germination rates and hyphal extension rates were measured under the microscope. The error bars represent standard error of the mean. Source data are provided as a Source Data file.

regulator of transcription, however, it is clear from the data presented in Fig. 3 that the NctA/B complex is responsible for regulating only a subset of genes at any one time. To assess if these subsets of genes are enriched for particular processes or biochemical pathways, GO term and Metabolic Pathway Enrichment analysis was performed on genes dysregulated more than twofold. Such analysis are somewhat limited when applied to *A. fumigatus* as GO terms and metabolic maps for filamentous fungi are poorly defined, however, we were able to identify enriched classes of genes in our upregulated cohort that are associated with secondary metabolism, transcriptional and translational processing, transport and notably steroid biosynthesis (Supplementary Fig. 6).

As both *nctA* and *nctB* null mutants are pan-azole resistant and our metabolic enrichment analysis identified steroid biosynthesis as an enriched class in genes upregulated in the null mutants, we assessed the mRNA levels of the genes in the ergosterol biosynthetic pathway (Fig. 3c, d)[40]. Of the 29 genes annotated as being associated with the pathway, 7 (*erg13A, hmg2, erg12,*

*erg11A, erg24A, erg6* and *erg2*) have increased mRNA levels more than 1.5-fold in the *nctA* null mutant (Fig. 3; Supplementary Data 2), while nine have reduced mRNA levels (*erg10A, erg13B, erg7C, erg24B, erg25A, erg26B, erg27, smt1* and *erg5*). Intriguingly for three of the downregulated genes, their paralogues are upregulated (*erg13B, erg24B, smt1* [paralog to *erg6*]). Notably, *erg7C* has recently been identified as an oxidosqualene:proto-stadienol cyclase which diverts 2,3-epoxysqualene from ergosterol biosynthesis into the helvoic acid biosynthetic pathway[41]. Potentially, reduced levels of *erg7C* would result in increased flux through the ergosterol biosynthetic pathway. Similarly, we observed significant downregulation of several siderophore biosynthesis genes in the *nctA* and the *nctB* null mutants, especially *sidI*, which acts to divert the ergosterol precursor mevalonate into the siderophore biosynthesis pathway[42] (Supplementary Data 2). Together with the upregulation of *hmg2* and *erg12* in the NCT mutants, this might suggest a potential role of the NCT complex in balancing ergosterol and siderophore biosynthesis.

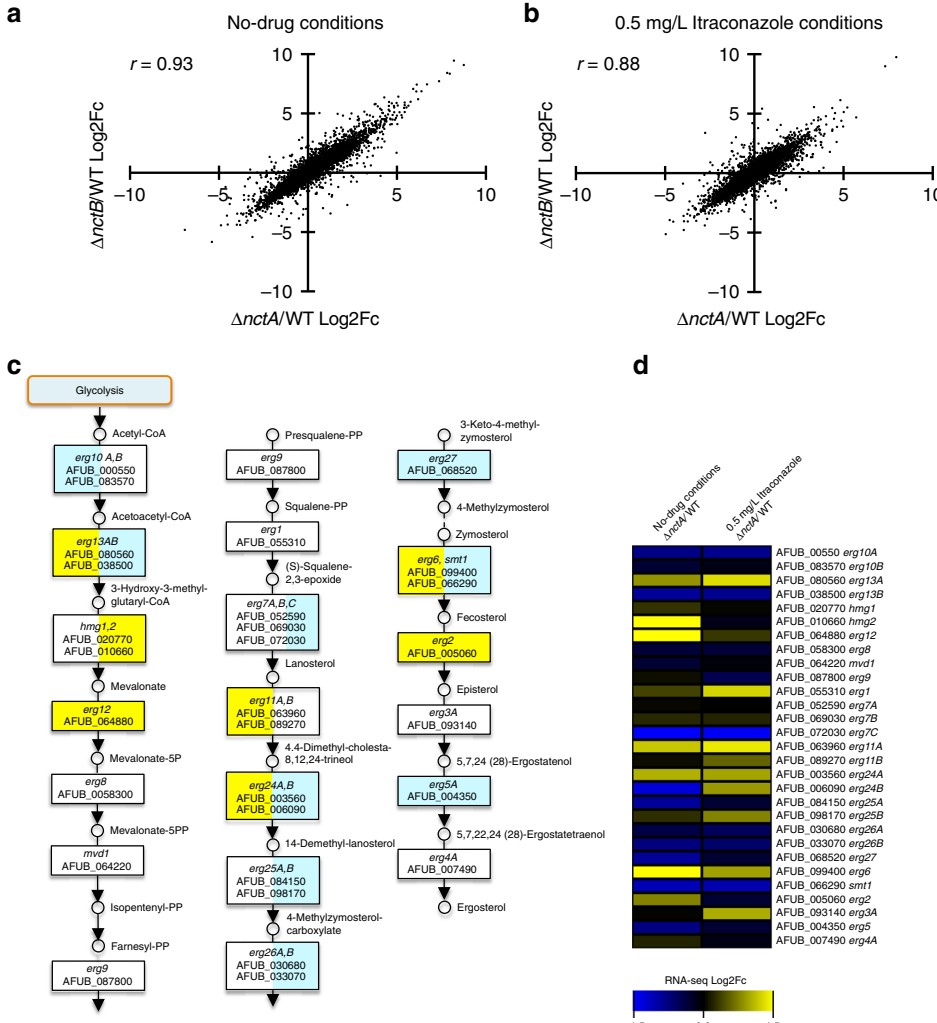

**Fig. 3 Effects of *nctA* and *nctB* deletion on the global and the ergosterol biosynthetic gene expression. a, b** Comparison of differential gene expression in the *nctA* and the *nctB* null mutants. Scatter plot comparison of the log2 differential expression ratio of gene expression data in (**a**) no-drug, and (**b**) 0.5 mg/L itraconazole conditions. The log2 expression ratio between the *nctA* null mutant and the wild-type are plotted on the *x*-axis, and the *nctB* null mutant and the wild-type are plotted on the *y*-axis. The Pearson's correlation (*r*) between the two gene expression data sets is shown. **c, d** Effects of *nctA* deletion on the expression of the genes involved in ergosterol biosynthesis. **c** The putative ergosterol biosynthetic pathway in *A. fumigatus*. The genes highlighted in yellow are those whose expression levels were upregulated more than 1.5-fold in the *nctA* null mutant compared with the wild-type. The genes highlighted in light blue are those whose expression levels were downregulated more than 1.5-fold in the *nctA* null mutant compared with the wild-type. **b** Heatmap showing the RNA-seq expression levels of the genes involved in ergosterol biosynthesis. Log2 differential expression values are scaled between −1.5 and 1.5, and displayed. Source data are provided as a Source Data file.

Our RNA-seq analysis indicates that the NCT complex is a key regulator of other transcription factors associated with azole resistance. Loss of *nctA* leads to upregulation of activators of the ergosterol biosynthetic pathway (2.2-fold for *srbA* and 1.6-fold for *atrR* in the no-drug conditions) and downregulation of the negative regulator-encoding gene *hapC* (approximately twofold, Supplementary Data 2). These results suggest that the NCT complex affects transcript levels of ergosterol biosynthesis genes both directly and indirectly by modulating the expression levels of these transcription factors.

**NCT binds the promoters of genes linked to azole susceptibility**. To identify the promoters of genes that are directly bound by the NCT complex, we performed chromatin-immunoprecipitation sequencing (ChIP-seq) using an anti-S-tag polyclonal antibody to isolate DNA bound to the NctA-S-tag fusion protein in vivo.

We identified 4811 and 4290 NctA-binding peak regions in the absence and the presence (0.5 mg/L) of itraconazole, respectively (*q*-value < 0.01, fold enrichment >1.5; Supplementary Data 3). A comparison of the ChIP-seq data sets showed that >70% of the peak regions are common between the conditions (Fig. 4a). Analysis of genome-wide occupancy of NctA revealed that the large majority (~70%) of NctA-binding peaks are located within 1.5 kb upstream of the translational start site (TSS) of an annotated gene (Fig. 4b). In total, 33.1% (3316 genes in no-drug conditions) and 29% (2945 genes in 0.5 mg/L itraconazole conditions) of the total predicted ORFs were assigned to have at least one NctA-binding event within their upstream region. Frequency distribution of NctA-binding peak summits showed that NctA is predominantly positioned around 300 bp upstream from the TSS (Fig. 4b). Applying the Multiple Em for Motif Elicitation (MEME) de novo motif discovery program[43], we identified several conserved nucleotide motifs within the ChIP peak regions (Fig. 4c). TATA-box like AT-rich motives were

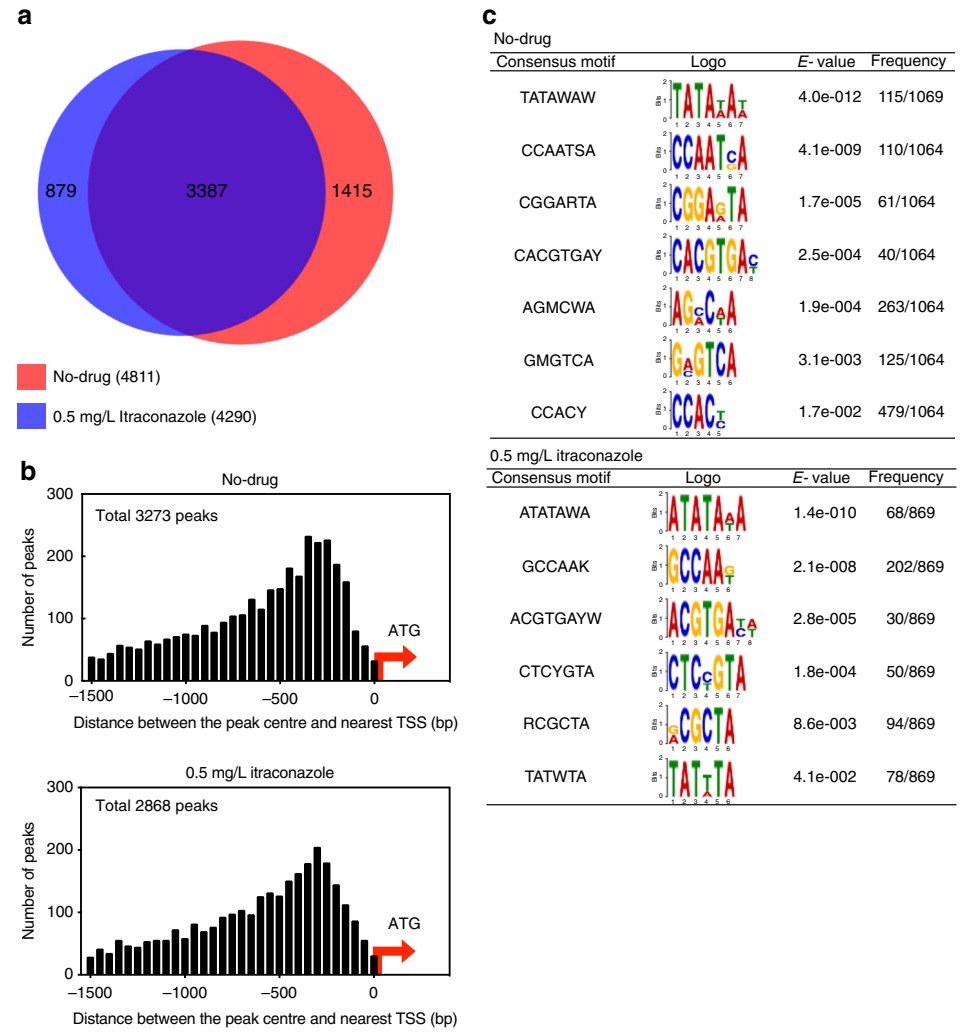

**Fig. 4 Genome-wide binding profile of NctA. a** Venn diagram showing the overlap of NctA-binding peaks (*q*-value < 0.01, fold enrichment >1.5) obtained from no-drug and the 0.5 mg/L itraconazole-treated conditions. **b** Distribution of the ChIP-seq peak location with respect to annotated genes. The frequency distribution of the distance between the merged peak summit of the significant binding peak regions (*q*-value < 0.01, fold enrichment >1.5) and the translation start site (ATG) of the nearest annotated gene is plotted. **c** Conserved nucleotide motifs identified in the NctA ChIP-seq peak regions. A summary of the de novo motif discovery analysis is shown with the identified consensus motifs and its sequence logos, the calculated *e*-values, and the frequency of the appearance of the motifs within the ChIP-seq peak set are analysed. Source data are provided as a Source Data file.

identified as the common nucleotide motifs with the highest *e*-value. This is consistent with the result from previous studies with NCT orthologues, which have been shown to form a stable complex with the TATA-box-binding protein (TBP)[44–47]. The results of the motif discovery suggest that the NCT complex in *A. fumigatus* also interacts with TBP to regulate gene expression as also suggested by the physical interaction studies.

We investigated the correlation between NctA occupancy and mRNA levels by comparing the ChIP-seq and the RNA-seq data sets. We expected that regulation of genes in close proximity to the NctA-binding sites would be altered upon loss of *nctA*. However, for the total 3361 genes having at least one NctA-binding peak within 1.5 kbp of upstream region, only 755 genes (22%) showed differential mRNA levels, and the remaining 2606 genes exhibited no significant change (Supplemental Fig. 7a, b). Among the 755 differentially expressed genes, 352 genes were upregulated and 403 genes were downregulated in the *nctA* null mutant in no-drug conditions. Similar results were obtained for the itraconazole-treated (0.5 mg/L) sample (Supplemental Fig. 7c, d). These results suggest that binding of the NCT complex alone is not sufficient for it to elicit its regulatory function.

In order to obtain further insight into the molecular mechanisms driving the itraconazole resistance in the *nctA* and the *nctB* null mutants, we analysed binding of NctA on the promoter region of the genes related to ergosterol biosynthesis and their known transcriptional regulators (Supplementary Data 2, Supplementary Fig. 8a). Of the 16 genes, which showed differential expression in the RNA-seq analysis in the absence of itraconazole, nine genes (*erg10A, erg13B, erg11A, erg24A, erg25A, erg27, erg6, smt1* and *erg5A*) were found to have at least one NctA-binding peak within 1.5 kb of the upstream region. Similarly, among the 14 differentially expressed genes in the presence of itraconazole, 11 genes were shown to have the peak regions within the defined upstream region (*erg10A, erg13A, erg13B, erg11A, erg11B, erg24A, erg24B, erg25B, erg6, smt1* and *erg3*). Interestingly, the genes encoding the regulatory proteins HapC, SrbA and AtrR, and Mot1 were also confirmed to have an NctA-binding peak in their upstream region (Supplementary Fig. 8b, Supplementary Data 2).

**Ergosterol levels are elevated in NCT complex mutants.** To analyse the effect of the loss of the NCT complex on ergosterol

biosynthesis, sterol levels were quantified using GC–MS. Consistent with our hypothesis, we observed a 60% increase in ergosterol content (w/w) in both mutant strains when compared with the isogenic control ($P < 0.04$; Fig. 5). Ergosterol levels in the reconstituted strain were indistinguishable from the wild-type ($P > 0.85$), and lower than observed in the knockout ($P = 0.0138$). Taken together, our data suggest that NctA and NctB are negative regulators of ergosterol biosynthesis via their interaction at the promoters of *cyp51A* and other genes that encode components of the ergosterol biosynthetic pathway.

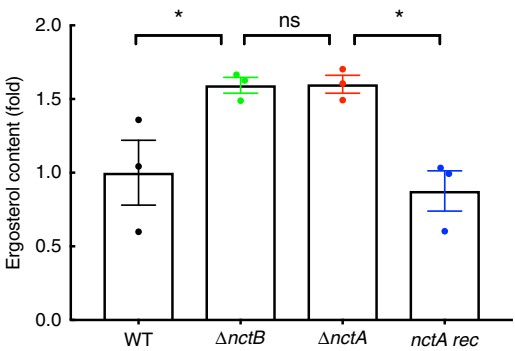

**Fig. 5 Loss of *nctA* and *nctB* increases the cellular ergosterol content.** Ergosterol levels of the wild-type (WT), the *nctA* null mutant (*ΔnctA*), the *nctB* null mutant (*ΔnctB*) and the *nctA* reconstituted isolate (*nctA rec*) in RPMI-1640 incubated for 24 h were determined by GC–MS. Ergosterol content of each mutant was normalised to that of the wild-type, and shown as a relative fold change. Samples were assessed in biological triplicates. *P*-values were calculated using one-way ANOVA: *$P < 0.05$; ns, $P > 0.05$. The error bars represent standard error of the mean. Source data are provided as a Source Data file.

**Levels of the azole transporter CDR1B are elevated in NCT mutants.** The relatively modest increase in expression of *cyp51A* (ca. 2.5-fold) and cellular ergosterol content (ca. 1.6-fold) in the *nctA* mutant did not appear consistent with the relatively large increase in itraconazole resistance (>64-fold), suggesting that factors independent of ergosterol biosynthesis may be influencing azole resistance in the NCT complex-deficient strains.

Recently, an association between azole resistance and increase in mRNA levels of the ABC transporter CDR1B has been described[30], and the transcriptional regulator AtrR has been shown to co-regulate both *cdr1B* and *cyp51A* expression[11]. We therefore examined our RNA-seq data to see if *cdr1B* was dysregulated in the *nctA* and *nctB* null mutants. mRNA levels of *cdr1B* were increased by 3.1 (FDR = $2.5 \times 10^2$) and 2.1 (FDR = 0.15)-fold, respectively, in the *nctA* and the *nctB* null (Fig. 6a). Furthermore, evaluation of our ChIP-seq data suggested that this regulation was related to a direct interaction between the NCT complex and the *cdr1B* promoter (Fig. 6b). To assess if this increase in transcript levels led to a concomitant increase in translated protein levels, we quantified levels of the transporter using an anti-CDR1B antibody[48]. In keeping with our transcriptomic results, CDR1B levels were increased in the *nctA* null mutant by >2.4-fold (Fig. 6c, d).

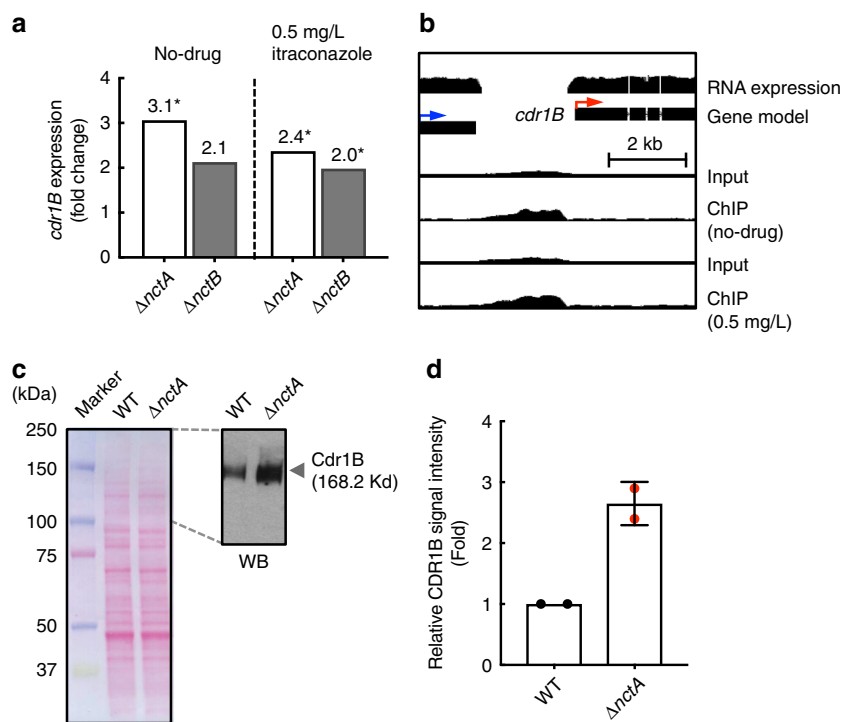

**Fig. 6 Defects in the NCT complex leads to transcriptional de-repression of *cdr1B* and over-production of CDR1B protein. a** Expression levels of *cdr1B* transcripts in the *nctA* null and the *nctB* null mutant in RNA-seq analysis: *, FDR < 0.05. **b** In vivo binding of NctA on the 5′-upstream region of *cdr1B*. Tracks for the NctA ChIP-seq (ChIP) and their input DNA control (Input) are visualised in the UCSC genome browser together with annotated gene models and their transcript, which are expressed in the no-drug conditions. The direction of the target gene and the 5′-proximal gene are shown in red arrows and blue arrows, respectively. **c** Representative western blot (WB) showing the increase in CDR1B protein levels in the *ΔnctA* mutant. Cell-free extracts were resolved via SDS-PAGE, and probed for CDR1B using a CDR1B-specific antibody. Ponceau S staining was performed as an overall loading control. **d** Relative protein levels of CDR1B. The relative intensity of the CDR1B signal was quantified by densitometric scanning. The data represent the mean results of two biological replicates, and the error bars signify the standard deviations. Source data are provided as a Source Data file.

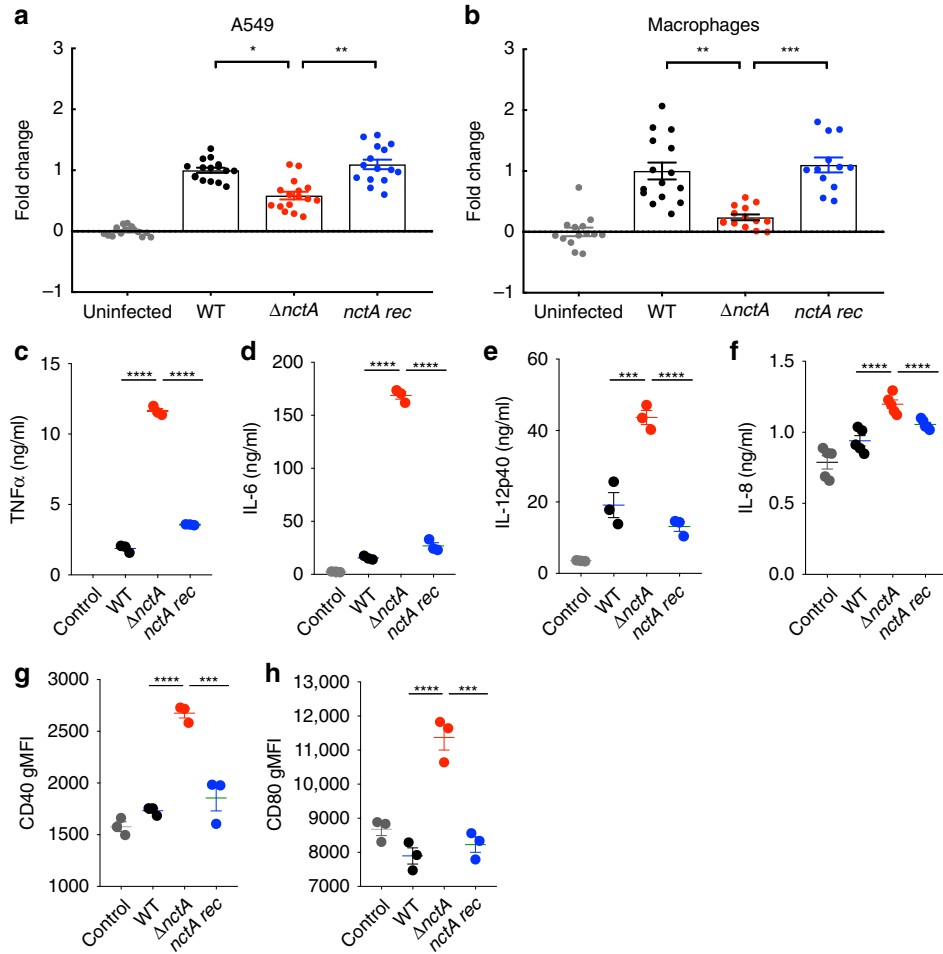

**Fig. 7 Cytotoxicity and immunogenic properties of the *nctA* null mutant. a**, **b** A549 epithelial cells or macrophages were infected with the wild-type (WT), the *nctA* null mutant (*ΔnctA*) or the *nctA* reconstituted isolate (*nctA rec*) for 24 h. Cytotoxicity of each mutant was evaluated by measuring the release of lactate dehydrogenase (LDH) activity into the culture medium. The data represent five different infection challenges with triplicate LDH activity measurements. Data are shown in fold change of LDH activity relative to the wild-type-infected cells. The error bars mean the standard error of the mean (SEM), and *P*-values were calculated by Kruskal–Wallis test with Dunn's correction: *\**P* < 0.0180; *\*\**P* < 0.0056 (for **a**, A549 cells). *\*\**P* < 0.0032; *\*\*\**P* < 0.0007 (for **b**, macrophages). **c**–**f** Granulocyte macrophage colony-stimulating factor-induced bone marrow-derived dendritic cells (gm-csf BMDCs) were infected with the *A. fumigatus* strains with the multiplicity of infection (MOI) = 5:1. Proinflammatory cytokines were quantified by ELISA. Data represent three biological replicates (**c**–**e**) or five biological replicates (**f**) with ± SEM. *P*-values were calculated by ANOVA with Tukey's correction: *\*\*\**P* < 0.002; *\*\*\*\**P* < 0.0001. **g**, **h** Activation of dendritic cells measured by CD40 and CD80 markers by flow cytometry. Data represent three biological replicates with ±SEM. *P*-values were calculated by ANOVA: *\*\*\**P* < 0.0002; *\*\*\*\**P* < 0.0001 (for **g**). *\*\*\**P* < 0.0004; *\*\*\*\**P* < 0.0001 (for **h**). Source data are provided as a Source Data file.

**The nctA null mutant shows enhanced immunogenic properties**. Given the requirement of the NCT complex in the resistance to two leading classes of therapeutic agents used to treat aspergillosis, it was important to assess if NCT complex-mediated regulation is important for pathogenicity of *A. fumigatus*. As our evidence to data suggests that the roles of NctA and NctB are non-redundant with respect to each other, we investigated pathogenic properties of the *nctA* mutants using both in vitro and in vivo infection models.

To assess contact-mediated cytotoxicity, the *nctA* null mutant was co-cultured with human A549 alveolar epithelial cells and the murine macrophage RAW 264.7 cell line (Fig. 7a, b). Our results indicate that loss of NCT complex results in a significant reduction in cell damage when compared with either isogenic control or the reconstituted isolate. This result is consistent with the growth reduction seen for this strain in the media used for this experiment (Fig. 2b; Supplementary Fig. 5c). We next examined immunogenic properties of the *nctA* null mutants to mammalian immune cells. Murine macrophages challenged with

live spores from the *nctA* mutant showed an increase in production of TNF-α, IL-6 and IL12p40 (Fig. 7c–e), whereas human epithelial cells challenged with the *nctA* mutant showed an increase in production of IL-8 (Fig. 7f). The *nctA* null mutant also resulted in more activation of dendritic cells compared with the wild-type (Fig. 7g, h). These findings suggest that the *nctA* null mutant could potentially cause an aberrant immunogenic response during infection. The increased immunogenic properties observed for the *nctA* null mutant could be associated with an alteration in the cell wall structure or spore surface, as the null mutant was hypersensitive to the cell wall synthesis inhibitory drugs and the perturbing agents Congo Red and Calcofluor White (Fig. 1c). However, examination by TEM revealed that the thickness of the cell wall was unchanged in the *nctA* null mutant (Supplementary Fig. 9).

**An NCT mutant is virulent in a model of invasive aspergillosis**. To assess the impact of the loss of the NCT complex in

pathogenicity, we compared the virulence of the *nctA* null mutant with that of the isogenic isolate MFIG001 in a leukopenic model of invasive aspergillosis. All mice challenged with the MFIG001 strain succumbed to infection (100% mortality) within 7 days post infection. Despite the significant growth defect (Fig. 2b) and the reduced cytotoxicity (Fig. 7a, b) observed in NCT mutants, the virulence of the *nctA* null mutant was statistically indistinguishable from the isogenic isolate (Fig. 8a).

In the leukopenic aspergillosis model, the host cellular innate response is quantitatively attenuated particularly during the initiation of infection allowing the microbe to proliferate and cause host damage. Thus, we investigated the role of the NCT complex in virulence in a murine model that uses cortisone acetate as an immunosuppressive agent resulting in qualitative defects in innate immunity. This model has been shown to better reflect the host's immune response in relation to detection of fungal-specific PAMPs. In this model, overall murine mortality with the isogenic strain was 45% at day 10 post infection. Similarly, the *nctA* null mutant also showed 50% mortality at the end of the infection time course. Although the *nctA rec* strain showed a slightly increased mortality compared with the other strains, no significant differences were observed in virulence between the all tested strains (Fig. 8b).

One potential hypothesis that could explain the retention of virulence of the *nctA* null mutant is the aberrant immunogenic properties observed in the mutant in vitro. However, no significant differences were observed in the expression levels of cytokine-encoding genes (TNF-α, IL12 and IL-6) between the strains at 36 h post inoculation (p.i.) (Fig. 8c). Furthermore, no obvious difference in the inflammatory response was observed between the strains in histopathology (Fig. 8d). Contrary to our expectation from the in vitro studies, the *nctA* null mutant showed an indistinguishable level of fungal burden with the isogenic control at 36 h p.i. (Fig. 8e). These results suggest that pathogenicity is maintained in this strain as its growth is not significantly altered in vivo, and our results add to the growing body of evidence that poor in vitro growth is not an absolute indicator of virulence defects[49,50].

## Discussion

In this study, we have constructed a 484 members transcription factor null mutant library with the aim of providing a systematic evaluation of regulators that contribute to azole resistance in *A. fumigatus*. This library now provides an opportunity for the fungal community to further explore regulatory mechanisms and factors in the pathobiology of this important human fungal pathogen. To date, much of our understanding of azole tolerance has been driven by hypotheses derived from model organisms that have significantly superior functional genomic resources. Functional screens of transcription factor null mutant libraries have been performed in the model yeasts *S. cerevisiae*[51] and *Schizosaccharomyces pombe*[52], the model filamentous fungus *Neurospora crassa*[53] and the pathogenic yeasts *Cryptococcus neoformans*[54] and *Candida albicans*[55]. Although these studies have proven to be effective in uncovering a number key regulators of drug resistance, the roles of transcription factors can vary significantly from species to species and large-scale transcriptional rewiring is frequently observed (for a review see ref. [56]). For example, loss of the *A. fumigatus* pH-responsive transcriptional regulator PacC, results in a >20-fold increase in flucytosine sensitivity in *A. fumigatus*[57], however, loss of the orthologue in yeast (RIM101) leads to an increase in flucytosine resistance[54,55]. Our screen of azole resistance in *A. fumigatus* highlights significant differences in the transcriptional regulation of azole tolerance in filamentous fungi compared with yeasts. Of the 12 transcription

factors identified in our screen as having altered susceptibility to itraconazole, only half HapB, SreA (SrbA)[54], AdaB[35,58], GisB[59] CreA and HapX[60] may have been predicted from previous screening efforts. Of the remaining regulators, functional orthologues of AreA exist in yeast (eg. Gat1p and Gln3p in *S. cerevisiae*), however, they have not been associated with changes in azole resistance while orthologues of three (NctA, NctB and RscE) are essential for viability, and two (ZipD and AtrR) are absent in yeasts.

We have explored, in detail, the role of the two CBF/NF-Y family transcription regulators, AFUB_029870 (NctA) and AFUB_045980 (NctB) in azole resistance. The orthologues of these regulators in *S. cerevisiae*, known, respectively, as Bur6 and Ncb2, are subunits of a heterotrimeric transcriptional regulator called Negative Cofactor 2 (NC2). The NC2 complex, originally identified as a TBP (TATA-box-binding protein)-associated factor, acts as a negative regulator of RNA polymerase II transcription by inhibiting formation of the pre-initiation complex (PIC)[61,62]. Assembly of the heteromeric PIC is required for transcription from RNA pol II-dependent promoters, and its assembly is contingent upon recruitment of the TBP and the general transcription factors TFIIA and TFIIB. NC2 inhibits PIC formation by preventing the interaction of TBP with TFIIA and TFIIB [ref. [63] and references therein]. The interaction between NCT and the TBP is regulated by the Swi2/Snf2-type ATPase transcriptional modulator Mot1[64]. Consistent with a role for Mot1 in regulating NCT–TBP complex binding, we have shown a direct physical interaction between Mot1 and NctA in *A. fumigatus*. Interestingly, it appears that the NCT complex also has a direct role in negatively regulating Mot1 activity as expression of *mot1* is upregulated in the *nctA* null mutants and binding sites for the NCT are found in the *mot1* promoter (Supplementary Data 2). In keeping with the role of the NCT as a general transcriptional cofactor, genome-wide binding studies in *S. cerevisiae*, *C. albicans* and *H. sapiens* have revealed interaction with in excess of 20% of all RNA pol II gene promoters[63–66]. These finding are consistent with our observation that NctA binds the promoters of over 30% of protein encoding genes of *A. fumigatus*, however, given this general role, what appears remarkable is that *nctA* and *nctB* are not essential for viability of this fungus as they are in yeasts[44] and loss of function mutants have clear and very specific phenotypic traits.

We have shown that loss of the NCT complex leads to a large increase in MIC to the azoles, most notably for posaconazole where we observed a shift in MIC from 0.25 mg/L to >16 mg/L. This phenotype is associated with transcriptional dysregulation of the ergosterol biosynthetic pathway directly and indirectly via *srbA*[10], *atrR*[11,15] and *hapC*[12], an increase in cellular ergosterol levels (c. 1.6-fold) and an increase in levels of the Cdr1B azole transporter (2.4-fold) (see model in Fig. 9). Although the increase in sterol levels in the NCT mutants is rather modest and apparently out of keeping with the large increase in MIC, recent evidence does not support a direct linear relationship between ergosterol levels and azole tolerance. Rybak et al. described mutation in the sterol sensing domain of *hmg1* ($hmg1^{F262del}$) that results in an apparent de-repression of ergosterol biosynthesis leading to a modest increase (1.6-fold) in ergosterol levels, but a much higher relative increase in MIC to itraconazole and isavuconazole (eight-fold)[67]. The reason for this incongruity is unclear, however, studies in *S. cerevisiae* have highlighted that changes in the composition of the plasma membrane affect the function of transporters. For example, depletion in sterol levels caused by loss of *erg4* or *erg6* leads to a reduction in the activity of the multidrug-resistance transporter Pdr5[68], and ergosterol is required to correctly localise the azole exporter Cdr1p in *C. albicans*[69]. This leads us to speculate that even relatively small increases in ergosterol content in the cell membrane may lead to large

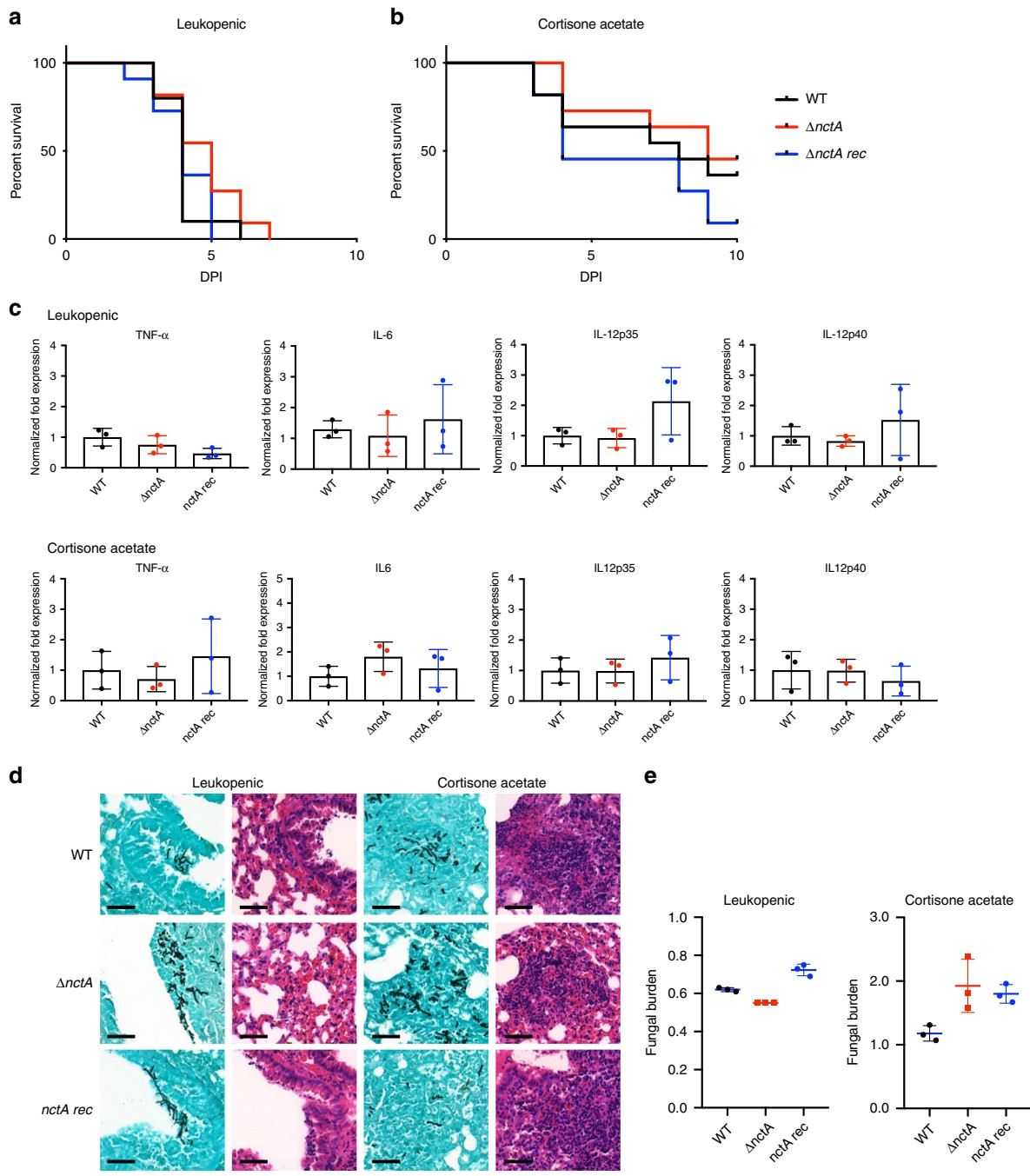

**Fig. 8 Assessing the effect of loss of *nctA* on the virulence of *A. fumigatus* in murine infection models.** Mice were infected with the wild-type (WT), the *nctA* null mutant (*ΔnctA*) or the reconstituted isolate (*nctA rec*). **a**, **b** Kaplan–Meier curve for murine survival for mice treated via (**a**) intranasal infection with $5.0 \times 10^5$ conidia after mice were rendered neutropenic by treatment with cyclophosphamide, and (**b**) intranasal infection with $7.0 \times 10^6$ conidia after cortisone acetate treatment. A log rank analysis was used to compare the results between the strains. **c** Relative expression levels of proinflammatory cytokines in the infected lung tissues. Data represent the mean of biological triplicates and error bars illustrate the standard deviation. Statistical significance was calculated by Student's *t* test. **d** Histopathology of representative sections of lungs after 36 h post infection with WT, *ΔnctA*, or *nctA rec*. Lung sections were stained with haematoxylin–eosin (H&E) for visualisation of the host cells, and Grocott's Methenamine Silver (GMS) for visualisation of the fungal elements. Scale bar: 50 μm. **e** Quantification of the fungal burden in the infected lungs. The total genomic DNA was extracted from the same lung lobe samples used for the cytokine expression analysis, and fungal DNA concentration was determined by qPCR. Data represent the means of biological replicates, and the error bars mean standard deviation. The statistical significance of variances between fungal burdens was calculated by using a non-parametric Mann–Whitney *t* test. Source data are provided as a Source Data file.

increases in azole resistance via an indirect effect on azole transporter levels. Given the pleiotropic nature of the NCT complex, we cannot exclude that further factors may be contributing to the high levels of azole resistance evident in the *nctA*

and *nctB* null mutants especially in light of our evidence showing that they are hypersensitive to the cell wall acting agents, and recent data showing a link between reductions in β-1,3-glucan synthesis and the delayed fungicidal effects of voriconazole[70].

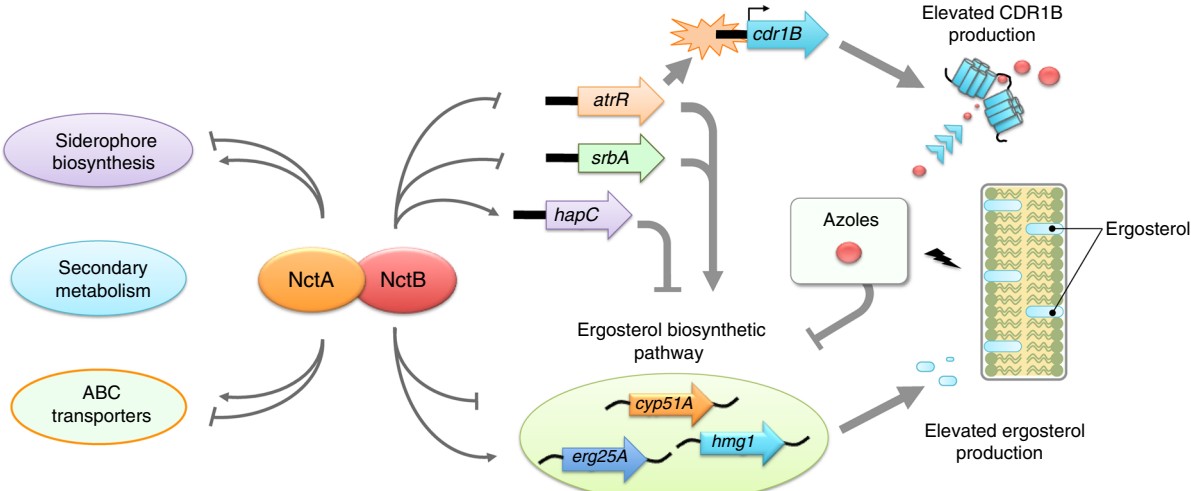

**Fig. 9 Proposed model highlighting the mechanistic basis of the azole resistance in the NCT complex.** The NCT complex is a global regulator, which acts as both a negative and a positive regulator of a wide range of gene regulation that includes secondary metabolism, cellular transport and sterol biosynthesis as important targets for virulence and azole resistance in *A. fumigatus*. The NCT complex fine-tunes expression of several genes in the ergosterol biosynthesis pathway and the azole efflux pump encoding *cdr1* by directly interacting with their core promoter region. The NCT complex also acts as a key regulator of azole resistance by modulating the expression levels of the transcription factors associated with ergosterol biosynthesis and azole resistance; transcriptional repression of the activator-encoding *srbA* and *atrR genes*, and activation of the negative regulator-encoding *hapC gene*. Therefore, the loss of the NCT complex causes an increased level of cellular ergosterol content and an elevated production of Cdr1B that leads to a multi-drug-resistance phenotype to the azoles as well as the salvage therapeutics amphotericin B and terbinafine.

The most striking phenotypes that we have observed for the *nctA* and the *nctB* null mutants aside from the resistance to the azole class of antifungals is their resistance to the salvage therapeutic amphotericin B and terbinafine, which can be used in the management of patients with chronic or allergic disease[19]. Cross-resistance to the azoles and terbinafine is understandable, as both act on the ergosterol biosynthetic pathway (terbinafine, inhibits the action of squalene epoxidase, an enzyme that catalyses the conversion of squalene to squalene 2,3-epoxide in the ergosterol biosynthesis pathway[71]). Cross-resistance between these agents and amphotericin B is much harder to explain. Amphotericin B acts by selectively binding ergosterol in the fungal membrane and creating pores resulting in leakage of intracellular contents[72]. Theoretically therefore, resistance to the azoles caused by upregulation of ergosterol in the cell membrane should lead to increased sensitivity to amphotericin B as this would enhance the interaction between the drug and its binding target. Indeed, this inverse correlation in resistance profiles has been observed in several mutants from *C. neoformans*[54] and in ergosterol biosynthetic mutants of *S. cerevisiae*[73]. Cross-resistance to the azoles and amphotericin B has previously been reported in artificially constructed yeast mutants. In *C. albicans*, laboratory generated strains lacking both copies of ERG11 (Cyp51A orthologue) or ERG3 (sterol 5,6, desaturase) are viable and resistant to fluconazole. As ergosterol is replaced in their cell membrane by alternative sterols, these isolates are also resistant to amphotericin B[74,75]. However, depletion of ergosterol levels is clearly not the key cause of amphotericin B resistance in the NCT mutants, as they exhibit an increase in ergosterol content in line with the observed increased expression of *cyp51A* and other genes of the ergosterol biosynthetic pathway. There is limited information on the mechanisms of amphotericin B resistance in *A. fumigatus*, however, it has been suggested that an increase in the production of oxidative stress-reducing enzymes such as catalases that confers resistance to the oxidising abilities of amphotericin B may contribute to resistance[75]. It is also possible that changes in the structure of the cell wall architecture in the *nctA/nctB* null strains is contributing to amphotericin B resistance by limiting access to

the ergosterol in the cell membrane. Clearly, additional studies are needed to further define the mechanism of amphotericin resistance in these strains. Our results gain greater significance as we have observed that, despite significant growth defects in vitro, the *nctA* mutant retained similar levels of virulence and growth in murine models of infection. This suggests that resistance observed in the *nctA/B* null mutants may be clinically relevant, however, as relatively few sequenced *A. fumigatus* isolates are in the public domain, a genetic association between drug resistance and polymorphisms in *nctA* and *nctB* is currently difficult to prove. Clinical guidelines currently suggest that amphotericin B may be used as a salvage therapeutic when initial therapy with voriconazole fails[18]. This would clearly be contraindicated in an infection with a strain with NctA/NctB-mediated azole resistance, so rapid detection of such isolates would support appropriate alternative therapeutic treatment such as an echinocandin to which these mutants are extremely sensitive (Fig. 1c).

The relevance of our findings for clinical drug resistance in pathogenic yeasts is unclear. Downregulation of Bur6 and Ncb2, in *C. albicans* leads to a modest increase in azole resistance, which has been attributed to a slight (ca. 2-fold) increase in *cdr1* expression[44]. To our knowledge however there have been no reports of amphotericin B resistance in these strains, and resistant clinical isolates carrying mutations in Bur6 and Ncb2 have not been reported. Critically, as both Ncb2 and Bur6 are essential for viability in *C. albicans*, null mutants are unlikely to persist in a host setting[44].

In summary, we have generated a library of transcription factor null mutants in the fungal pathogen *A. fumigatus*. This library is publically available and can be exploited by the research community to provide comprehensive insights into transcriptional networks governing critical factors associated with the cell biology, pathogenicity and allergenicity in this understudied aetiological agent. Using this resource, we have identified the network of regulators governing azole resistance and identified a novel mechanism that, through a single genetic mutation, is able to drive both high level pan-azole resistance and cross-resistance to the salvage therapeutic amphotericin B and terbinafine without significantly impacting virulence.

## Methods

**In silico identification of transcriptional factors.** Putative transcription factors were identified by through a combinatorial approach. Semantic searches using the terms transcription factor, C6, C2H2, HLH, transcriptional regulator, zinc finger, bZIP, CCAAT, CBF, MYB, DNA binding and TATA were performed to identify genes annotated as transcription factors within databases at ENSEMBL fungi and ASPGD. These were combined with genes annotated as transcription factors at the DBD transcription factor database. Duplicates were removed and each gene was manually curated to remove those lacking functional domains associated with transcription factor function (as deemed by searches for PROSITE, SUPER-FAMILY, SMART or CDD domains) or those that lack functional data indicating a direct role in transactional regulation.

**Generation and validation of transcription factor null mutants.** The transcription factor null mutant collection was generated in the *A. fumigatus* strain MFIG001 (previously known as A1160 *Δku80 pyrG+*)[30]. Gene replacement cassettes were generated using a fusion PCR approach (see Supplementary Fig. 2)[29]. Briefly, primers P1 and P2 were used to amplify around 1 kb of the 5′ flank, while P3 and P4 were used to amplify the 3′ flank. Primers hph_F and hph_R were used to amplify a 2.8 -kb hygromycin B phosphotransferase cassette from pAN7-1. PCR products were purified by solid phase extraction with the Qiagen QIAquick® PCR purification kit (Qiagen). Fusion of the three products was facilitated by the presence of common linker sequences on primers P2 and hph_F, P3 and hph_R and the use of the nested primers P5 and P6. PCR amplification was carried out using the protocols defined by Szewczyk et al.[29] Transformation was carried out as previously reported[30]. The sequences of all primers used are given in Supplementary Data 1.

Validation of homologous recombination and single integration of the deletion cassette was performed by PCR (Supplementary Fig. 2). PhusionFlash High-Fidelity Master mix (ThermoFisher Scientific) was used for all reactions. Primers P1 with hph-chk 5′-Rv and hph-chk 3′-Fw with P4 were used to amplify a region of about 1 kb from within the deletion cassette to the flanking region outside of the deletion cassette. Furthermore, PCR was performed with P1 and P4 as primers to check the purity of the gene knockout strain.

**Drug sensitivity screening.** Conidia of 484 TFKO strains were inoculated in 25 -mL culture flasks containing ACM + 100 μM hygromycin. Conidia were harvested by filtration and counted via optical density measurements. Approximately 2000 spores were inoculated per well of a CytoOne® 96-well plate (StarLab) containing RPMI-1640 medium 2.0% glucose and 165 mM MOPS buffer (pH 7.0) with 0, 0.06, 0.12 or 0.5 mg/L itraconazole. Plates were incubated at 37 °C for 48 h (0, 0.06, 0.12 mg/L or 96 h (0.5 mg/L), and optical density measurements were taken at 600 nm. Fitness was calculated by normalising optical density to the wild-type strain. Relative fitness was obtained by normalising fitness under itraconazole challenge to fitness of this strain under no itraconazole challenge. MIC determination for all drugs was carried out according to methods outlined by EUCAST[32]. General growth fitness of the transcription factor nul mutants in liquid RPMI-1640 medium was determined in the same microculture conditions ($n = 3$). Relative fitness was obtained by optical density measurements ($OD_{600}$) normalised to the wild-type strain.

**Radial growth germination rate and hyphal extension analysis.** Radial growth of the wild-type strain (MFIG001), the *nctA* null mutant or the *nctA* reconstituted strains was measured by inoculating 500 spores on *Aspergillus* Minimal Media (AMM), *Aspergillus* Complete Media (ACM), RPMI-1640 medium with 2.0% glucose and 165 mM MOPS buffer (pH 7.0) or Dulbecco's Modified Eagle's Medium (DMEM) on petri dishes. Plates were incubated at 37 °C for 72 h, and the radius of colonies was measured every 24 h.

Germination rate and hyphal extension rate were determined as follows. In total, 500 μl of $5 × 10^5$ spores of the strains were inoculated in the RPMI-1640 medium containing 2.0% glucose and 165 mM MOPS buffer (pH 7.0) in a 24-well glass bottom plate. The culture was incubated at 37 °C, and either optical density (600 nm) was measured on a Synergy 2 Multidetection Microplate reader (BioTek) or images were taken on a Leica SP8X confocal microspoce (Leica). Sporulation and hyphal length were measured in ImageJ.

**S-tag co-immunoprecipitation.** C-terminal S-tagged cassettes were generated by fusion PCR using primers SP1-SP8 (Supplementary Data 1). Three separate PCR reactions were performed for initial amplification of the cassette components. Primers SP1 and SP2 were used to amplify the 5′ flank and TF coding sequence, primers SP3 and SP4 amplified the downstream region of the TF (ca. 1 kb) from MFIG001 genomic DNA. Primers SP5 and SP6 were used to amplify the S-Tag, G5A linker and *pyrG* gene from pHL81[76] PCR products were purified by solid phase extraction with the QIAquick® PCR purification kit (Qiagen). Fusion of the three products was facilitated by the presence of linker sequences on primers SP2, SP3, SP5 and SP6 and the use of nested primers SP7 and SP8. The cassette was transformed into *A. fumigatus* A1160p- and NctA-S-tag and NctB-S-tag strains were validated by PCR as described above for the gene KO process. Proteins were extracted from 16 h shake flask cultures (SAB medium, incubated at 37 °C). Briefly,

biomass was frozen in liquid nitrogen and ground to a fine powder before incubation on ice in 6 ml of ice-cold HK buffer (100 mM NaCl) per 1 g biomass for 30 min. Samples were centrifuged at $7500 × g$ for 30 min, and filtered through glass wool to remove cellular debris. To purify the S-tagged proteins, crude protein extract was incubated with S-protein agarose beads (Novagen) with gentle agitation at 4 °C for 2 h. Samples were centrifuged, and the agarose bead pellet was washed in ice-cold HK Buffer with 100 mM NaCl and transferred to a S-Protein spin column (Novagen), where the beads were washed a further six times with 700 μl HK Buffer with 100 mM NaCl. S-tagged proteins were eluted using 50 μl Laemmli sample buffer. Eluted proteins were analysed by the Protein Mass Spectrometry at the Biological Mass Spectrometry Core Facility at the University of Manchester.

**Chromatin immunoprecipitation (ChIP) of S-tagged NctA.** In total, $1 × 10^6$ spores/ml of the wild-type strain (MFIG001) or the S-tagged NctA expressing strain were grown in 50 ml of Vogel's minimal medium for 18 h at 37 °C with constant shaking at 180 rpm. The mycelia were harvested by filtration, washed twice with distilled water and were transferred into 50 ml of RPMI-1640 medium (Sigma-Aldrich) containing 2.0% glucose and 165 mM MOPS buffer (pH 7.0). The cells were incubated for 4 h in the absence and the presence (0.5 mg/l) of itraconazole at 37 °C under shaking. Cross-linking was carried out by the addition of formaldehyde to a final concentration of 1.0% followed by incubation at 37 °C for 20 min. The cross-linked mycelia were ground to a fine powder under liquid nitrogen, and ~100 mg of the mycelial powder was suspended in 1.0 ml of ChIP lysis buffer (50 mM HEPES pH 7.5, 150 mM NaCl, 1 mM EDTA, 1% Triton X-100, 0.1% deoxycholate (Sigma D6750), 0.1% SDS, 1 mM PMSF and fungal proteinase inhibitor cocktail (Sigma)). The cell suspension was then sonicated with a Q125 sonicator (Qsonica, USA) to shear the chromatin to fragments with an average size of 0.2–0.5 kbp. After sonication, the insoluble cell debris was pelleted by centrifugation, and the soluble fraction was used for ChIP experiments and input control preparations. ChIP reaction was performed with an Anti-S-tag polyclonal antibody (ab18588, abcam) on Dynabeads Protein A magnetic beads (ThermoFischerScientific). Immunoprecipitated DNA was reverse cross-linked, treated with RNaseA (Sigma-Aldrich) and then purified using a MinElute PCR purification kit (Qiagen). To prepare input control, 100 μL of the sonicated extract was reverse cross-linked, treated with RNaseA (Sigma-Aldrich) and then purified using a MinElute PCR purification kit (Qiagen).

**ChIP-sequencing analysis.** ChIP-seq libraries were constructed following the manufacturers instructions for Illumina ChIP-seq library preparation. Eight samples were indexed and sequenced in a single lane on the Illumina HiSeq2500 as paired-end reads.

Raw sequencing reads were quality controlled with Illumina chastity filter and fastqc v0.11.3, and then Illumina adapters were trimmed from them using Trimmomatic. The resulting reads were aligned to the *A. fumigatus* A1163 CADRE genome from Ensemble fungi, version 26 using Bowtie2. Peak calling was carried out using a Model-based Analysis for ChIP-Sequencing (MACS2[77]) version 2.1.0 with a *q*-value cut-off of 0.01. The results reported herein are for the combined reads from two biological replicate samples. All ChIP-seq experiments were carried out in two biological replicate samples. All ChIP-seq data sets are deposited in the NCBI Gene Expression Omnibus (GEO; https://www.ncbi.nlm.nih.gov/geo/) under accession number GSE129967 in a GEO Superseries GSE133491.

**Conserved motif discovery.** The reproducible 100 bp of the merged ChIP peak regions with more than twofold peak enrichment compared to the corresponding were analysed for conserved nucleotide motifs using the MEME suite[78] version 4.12.0. DNA sequence of the ChIP peak regions was retrieved from the genomic locations using the getfasta function from the BED tools suite[79] and analysed with the default setting.

**Transcriptomic analysis.** In total, $1 × 10^6$ spores/ml of the wild-type (MFIG001), the *nctA* null or the *nctB* null mutant was grown in 50 ml of Vogel's minimal medium containing 1.0% glucose for 18 h at 37 °C on a rotary shaker (180 rpm). Mycelia were collected by filtration, and washed twice with distilled water. About 1.0 g of wet mycelia were transferred into 50 ml of RPMI-1640 medium containing 2.0% glucose and 165 mM MOPS buffer (pH 7.0), and the cells were incubated for 4 h in the absence and the presence of itraconazole (0.5 mg/mL) at 37 °C with shaking. The drug-treated mycelia were then collected by filtration, immediately frozen with liquid nitrogen and kept at −80 °C until use.

The total RNA was extracted using TRI Reagent® (Sigma-Aldrich) according to the manufacture's instructions. The extracted RNA samples were treated with RQ1 RNase-Free DNase (Promega) and further purified using the RNeasy Mini Kit (Qiagen).

For paired-end RNA sequencing, libraries were generated using the TruSeq® Stranded mRNA assay (Illumina, Inc.), according to the manufacturer's instructions. Eight samples were indexed and sequenced in a single lane on the Illumina HiSeq2500. Generated Fastq files were analysed with FastQC, and any low-quality reads were trimmed with Trimmomatic. All libraries were aligned to the *A. fumigatus* A1163 genome assembly GCA_000150145.1 with the gene annotation from CADRE/Ensembl Fungi v24 using Bowtie, and only matches with

the best score were reported for each read. All RNA-seq experiments were carried out in three biological replicates.

Differential expression analysis was performed using DESeq[80].

All RNA-seq data sets are available in the NCBI Gene Expression Omnibus under accession number GSE133464 in a GEO Superseries GSE133491.

**Gene set enrichment analysis**. Gene ontology, functional category and KEGG pathway enrichment analysis were carried out using FungiFun2 2.2.8 BETA[81] web-based server (https://elbe.hki-jena.de/fungifun/fungifun.php) with the *A. fumigatus* A1163 genome annotation. Differentially expressed genes showing more than twofold enrichment with FDR < 0.05 were subjected to the enrichment analysis. Significance level of the enrichment was analysed using the Benjamini-Hochberg adjustment method with a *P*-value cut-off <0.05.

**Sterol analysis**. Conidia ($1 \times 10^6$) were grown in 50 mL RPMI-1640, incubated for 24 h at 37 °C with shaking (200 rpm). Mycelia were harvested, freeze dried and dry weights obtained prior to processing. Pellets were sonicated thoroughly ($6 \times 30$ s with Branson Digital Sonifier 250) in 1 ml of ddH$_2$O, and an internal standard of 10 µg of cholesterol was added to each sample.

Sterols were extracted and derivatised as previously described[82]. Briefly, lipids were saponified using alcoholic KOH and non-saponifiable lipids extracted with hexane. Samples were dried in a vacuum centrifuge and were derivatised by the addition of 0.1 ml BSTFA TMCS (99:1, Sigma-Aldrich) and 0.3 ml anhydrous pyridine (Sigma-Aldrich) and heating at 80 °C for 2 h. TMS-derivatised sterols were analysed and identified using GC/MS (Thermo 1300 GC coupled to a Thermo ISQ mass spectrometer, ThermoFischerScientific) and Xcalibur software (ThermoFischerScientific). The retention times and fragmentation spectra for known standards were used to identify sterols. The quantity of ergosterol (w/w) was calculated using the peak areas of ergosterol and the cholesterol internal standard from triplicate samples.

**Western blotting of CDR1B**. In total, $1 \times 10^6$ of *A. fumigatus* conidia were inoculated in a petri dish containing 20 mL of Sabouraud dextrose broth at 37 °C for 16 h. Mycelium that formed as a biofilm on the top was collected (100 mg), and was ground into fine powder in liquid nitrogen using a mortar and pestle. The ground mycelium was resuspended in 0.5 ml of 10% trichloroacetic acid (TCA), thoroughly vortexed, incubated for 5 min at room temperature and then washed three times in 90% acetone with 20 mM HCl and air dried. The TCA precipitate was then extracted with 8 M urea sample buffer (8 M urea, 5% SDS, 1% β-ME, 40 mM Tris-HCl (pH 8.0), with traces of bromophenol blue). Aliquots of this suspension were electrophoresed by 10% SDS-PAGE and then transferred to a nitrocellulose membrane. The membrane was blocked with 5% non-fat dry milk in phosphate-buffered saline with 0.1% Tween 20, and then probed with a polyclonal rabbit antiserum. The rabbit anti-CDR1B polyclonal antibody[48] was used with a horseradish peroxidase-conjugated secondary antibody (NA934V, GE Healthcare) and a SuperSignal West Pico chemiluminescent substrate (ThermoFisher Scientific) to visualise imunoreactive protein on an X-ray film.

**Murine infection models**. The murine infection experiments were performed under UK Home office Project Licence PDF8402B7 and approved by the University of Manchester Ethics Committee.

*A. fumigatus* strains were cultured on ACM containing 5 mM ammonium tartrate for 6 days at 37 °C, and conidia were harvested in sterile saline and used for infection experiments.

CD1 male mice (Charles River UK, Ltd.) were housed in groups of 3–4 in IVC cages with access to food and water ab libitum. All mice were given 2 g/L neomycine sulphate in their drinking water throughout the course of the study. For the leukopenic model of infection, mice were rendered leukopenic by administration of cyclophosphamide (150 mg/kg of body weight; intraperitoneal) on days −3, −1, +2 and every subsequent third day, and a single subcutaneous dose of cortisone acetate (250 mg/kg) was administrated on day −1. For the cortisone acetate model, mice were immunosuppressed with cortisone acetate (250 mg/kg), which were administrated subcutaneously on days −3, −1, +2 and every subsequent third day. Mice were anaesthetised by exposure to 2–3% inhalational isoflurane and infected by intranasally with a spore suspension of $1.25 \times 10^7$ conidia/ml (leukopenic model) or $1.75 \times 10^8$conidia/ml (cortisone acetate model) in 40 µl of saline solution. Mice were weighed every 24 h from day −3, relative to day of infection, and visual inspections were made twice daily. In the majority of cases, the endpoint for survival in experimentation was a 20% reduction in body weight measured from day of infection, at which point mice were killed. Kaplan–Meier survival analysis was used to create a population survival curve and to estimate survival over time, and *p*-values were calculated through a log rank analysis.

**Histology analysis**. Immunosuppressed male CD1 mice ($n = 3$) were infected as described above. After 36 h of infection, mice were killed and lungs were partitioned into lobes destined for histology analysis or fungal burden and cytokine mRNA expression analysis. Lobes for fungal burden and cytokine analysis were snap-frozen in liquid nitrogen and stored at −80 °C until use. Lobes for histological

analysis were immediately fixed in 4.0% (v/v) formaldehyde (Sigma-Aldrich), and subsequently embedded in paraffin. Four-micrometer sections were stained with haematoxylin–eosin (H&E) and Grocott's Methenamine Silver (G.M.S). Images were taken using a Pannoramic 250 Flash Slide Scanner (3D HISTECH) using brightfield illumination.

**Fungal burden analysis**. The total genomic DNA was extracted from the infected lung samples using a standard CTAB DNA extraction method. Fungal burden was determined by quantitative real-time PCR (qPCR) as described previously[83]. qPCR was performed in a 7500 Fast Real-Time PCR system (Applied Biosystems) using the SsoAdvanced Universal SYBR Green Supermix (Bio-Rad) with the primers listed in the Supplementary Dataset 1. Amplification reactions were performed in triplicate in a final volume of 20 µL using 0.5 µM forward primer, 0.5 µM reverse primer and 100 ng of total DNA. qPCR data were analysed as described[83]. The statistical significance of variances between fungal burdens was calculated by using a non-parametric Mann–Whitney *t* test.

**Cytokine expression analysis**. The total RNA was extracted from the infected lung samples using the RNeasy kit (Qiagen). First-strand cDNA was synthesised using the iScript cDNA synthesis kit (Bio-Rad) with 500 ng of the total RNA as a template. Amplification reactions were performed using the iTaq Universal SYBR Green Supermix (Bio-Rad) in a final volume of 20 µL using 0.5 µM forward primer, 0.5 µM reverse primer and 2 µL of fivefold diluted cDNA. Primers used in this analysis are listed in Supplementary Dataset 1. Relative expression level of gene expression was analysed using the ΔΔCt method with the murine actin encoding *actB* as the reference. Experiments were performed in biological triplicates. A two-sided Student's *t* test was used for statistical analysis, where *P*-value of <0.05 were considered as significant.

**Cell toxicity assays**. A549 human pulmonary carcinoma epithelial cells (American type culture collection, CCL-185) and Raw 264.7 macrophages were used under passage 20. Cells were maintained at 37 °C, 5% CO$_2$ in Dulbecco's Modified Eagle's Medium (DMEM), 10% foetal bovine serum (FBS) and 1% penicillin–streptomycin (Sigma-Aldrich). For all experiments, $2 \times 10^5$ A549 or RAW 264.7 cells were seeded in 24-well plates and incubated for 16 h when confluence equals 90%. Cells were then challenged with $10^5$ spores of Δ*nctA*, *nctA rec* and the isogenic control and incubated for 24 h. Following co-incubation with *A. fumigatus* spores, cell culture supernatants were collected, and the level of inflammatory markers or cell toxicity via lactate dehydrogenase assay were measured (Promega).

The concentration of IL-8 and IL-6 were determined in A549 epithelial cells co-cultured with *A. fumigatus* strains by using the Human IL-8/CXCL8 and IL-6 DuoSet ELISA according to the manufacturer's instructions (R&D systems). Experiments were performed in five biological replicates and technical triplicates. For data analysis, a four-parameter logistic (4-PL) curve was created plotting the absorbance versus Log$_{10}$ concentration of the standards and then sample concentrations determined by using a non-linear regression. Differences in IL-8 and IL-6 concentration between A549 cells challenged with Δ*nctA*, *nctA rec* and the isogenic control spores and uninfected controls were determined by one-way multiparametric ANOVA with Dunnet's correction using GraphPad Prism 7.0 (La Jolla, CA, USA).

**BMDC culture**. GM-CSF induced BMDCs were generated as previously described[84]. Briefly, bone marrow cells from C57BL/6 mice were seeded at $2 \times 10^5$/mL in complete media (RPMI-1640 (Sigma) plus 20 ng/mL GM-CSF (Peprotech), 10% FCS, 2 mM L-glutamine (Gibco), 100 U/mL penicillin 100 µg/mL streptomycin (Sigma)). Cells were cultured for 10 days, with 50% of the media replaced on days 3, 6 and 8. On day 10, DCs were re-plated at $2 \times 10^5$ cells/well, with $10 \times 10^5$ *A. fumigatus* spores (MOI 5:1), and incubated for 6 h at 37 °C.

**Flow cytometry and ELISA**. Post incubation, cells were taken for flow cytometry, and supernatants for ELISA. Cells were plated at $1 \times 10^6$ cells/well, washed twice in PBS, and stained with Zombie UV (Biolegend). FcR block (Biolegend) was then added, in addition to the following antibodies: CD11c-APCef780, CD40-PE CD80-PerCP/Cy5.5 (all Biolegend). Cells were washed twice in flow buffer (PBS 2 mM EDTA (Sigma) 2% FCS (Sigma)), and samples were acquired on a BD Fortessa, and analysed using Flowjo v10 (TreeStar). Cytokines for ELISA were measured using purified coating, detection antibodies and standards (Biolegend) or duosets (R&D), as per the manufacturers protocol.

**TEM imaging**. For transmission electron microscopy, the samples were fixed with 4% formaldehyde + 2.5% glutaraldehyde in 0.1 M HEPES buffer (pH 7.2). Then samples were incubated in 1% sodium met-periodate (in H$_2$O) for 1 h. After that they were postfixed with 1% osmium tetroxide + 1.5% potassium ferrocynaide in 0.1 M cacodylate buffer (pH 7.2) for 1 h and finally in 1% uranyl acetate in water for 1 h. Specimens were dehydrated in ethanol series infiltrated with TAAB Low Viscosity resin and polymerised for 24 h at 60 °C. Sections were cut with Reichert Ultracut ultramicrotome and observed with FEI Tecnai 12 Biotwin microscope at

100 kV accelerating voltage. Images were taken with Gatan Orius SC1000 CCD camera.

**Reporting summary**. Further information on research design is available in the Nature Research Reporting Summary linked to this article.

## Data availability

All RNA-seq and ChIP-seq data sets are available in the NCBI Gene Expression Omnibus (GEO) under accession number GSE133491. This GEO Superseries includes two SubSeries for the RNA-seq dataset (GSE133464 and the ChIP-seq dataset [GSE129967]. A processed format of the RNA-seq and the ChIP-seq data sets is included in Supplementary Data 2 (RNA-seq) and 3 (ChIP-seq). The source data underlying Figs. 1a-c, 2b-d, 3a-b and d, 4b, 5, 6c-d, 7a-h, 8a-c and e, and Supplementary Figs 1a, b, 3c, 5c–e, 6a–d, and 9 are provided as a Source Data file.

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

## Acknowledgements

The authors would like to thank the Genomic Technology Core Facility, the Bioinformatics Core Facility, the Biological Mass Spectrometry Facility, the Bioimaging Facility, the Flow cytometry Facility, and the Electron Microscope (EM) core Facility in the University of Manchester for their technical support. The EM core facility is supported by the Wellcome Trust equipment grant. We would also like to thank Dr Ian Donaldson of the Bioinformatis Core Facility at the University of Manchester for providing support analysing the ChIP-seq data, Dr Gareth Howell of the Flow cytometry Core Facility the University of Manchester for his assistance in conducting FACS analysis, Dr. Darren Thomson at the for his support in microscopy experiments. This work was supported by the Medical Research Council (MRC) grant MR/M02010X/1 to M.B., P.B. and E.B., the Wellcome trust grant 208396/Z/17/Z to P.B. and M.B. and by NIH R01AI143198-01 to M.B., P.B. and W.S.M. S.G. was co-funded by the NIHR Manchester Biomedical Research Centre and a NC3Rs Training Fellowship (NC/P002390/1). R.A.C. holds an Investigator in the Pathogenesis of Infectious Diseases Award supported by the Burroughs Wellcome Fund (BWF) and is also supported by a National Institute of Allergy and Infectious Diseases (NIAID) award 2R01AI081838.

## Author contributions

M.B., P.B., E.B., J.P.L., T.F., N.V.R., M.F., F.G. and E.D. conceived and designed the experiments. M.B., T.F., N.V.R., M.F., J.G., F.G., E.D., S.G., P.C., R.F.G., S.R., E.H., C.K., S.R., S.P. and J.E.P. performed experiments. T.F., N.V.R., S.R., J.M.B., J.G., E.H., C.K., J.E.P., S.K., R.A.C., P.C., S.M.R., E.B., P.B. and M.B. performed data analysis. R.F.G., J.E.P., S.K., R.A.C., J.P.L., P.C., S.M.R. and E.B. contributed reagents, materials, and analysis tools. T.F., N.V.R., P.B. and M.B. prepared the paper.

## Competing interests

M.B. is a consultant to Synlab GmbH and is the director and shareholder of Syngenics Limited. The remaining authors declare no competing interests.
