## [Peer Review File · Nature Communications]

Reviewers' comments:

Reviewer #1 (Remarks to the Author):

In this interesting and in part surprising manuscript, the authors describe their extensive efforts to unravel transcriptional regulation of relevant traits of the human pathogenic mould *Aspergillus fumigatus* on a global scale. Starting from the Hercules task of generating a comprehensive mutant strain library by targeting annotated transcription factor-encoding genes, the focus was set on means of resistance towards azole class antifungals. By this, it could be revealed that the Negative Cofactor 2 (NC2) orthologue of *A. fumigatus* might serve as master regulator of ergosterol biosynthesis and beyond: corresponding mutants display elevated immunogenicity but unaltered virulence in two murine infection models of invasive pulmonary aspergillosis.

While the general aspect of describing a valuable resource for the first time is highly appreciated, the subsequent tasks are presented to some degree imprecisely to leave some questions that might be addressed (see comments below). Accordingly, several minor points of criticism exist that ought to be revised in a future version of this highly relevant piece of work that doubtlessly will be of high value not only for the fungal research community.

In detail, specific issues are as follows:

The Introduction would benefit from a concluding sentence summarizing the relevance of this very study.

Why exactly does the lack of understanding azole resistance mechanisms prevent the development of combination therapies (l. 88-90)?

In the Results, the strategy to identify TF-encoding genes is unclear: given that for 84 proteins no functional domain could be identified (l. 153/4), why were these included in the first place?

The gene targeting approach using NHEJ-deficient *A. fumigatus* recipients is incompletely referenced (l. 160).

When selecting 12 mutants for genotype confirmation by Southern hybridization (l. 169), were any of the subsequently used strains among these? And were the genotypes of all the strains deleted for key regulators associated with azole resistance and sensitivity confirmed by this approach?

What is exactly meant by the term "higher level transcription factors" (l. 172/3) and why might the encoding genes be essential for *A. fumigatus*?

The fact that two master regulators of catabolite repression – CreA and AreA – appear to influence azole resistance (l. 184 and l. 189) raises the question on which type of culture medium and in the presence of which sources of carbon and nitrogen the corresponding assays were performed.

The general fitness (and its definition) of the key regulator deletants should be briefly described before elaborating on their phenotypes with respect to azoles or other drugs or cell wall stressors. What is actually the MIC for the recipient isolate MFIG001 that determines the sub-MIC and above MIC levels that were used for screening the transcription factor null library?

When characterizing the CBF/NF-Y transcription factors NctA (is its paralogue AFUB_045980 also part of the knock-out library?) and NctB, why was the focus set on the *nctA*Δ strain, to what degrees were phenotypes displayed by the *nctA*Δ and *nctB*Δ mutants similar?

The description and comparison of the RNAseq datasets (l. 224-227) lacks the one in presence of itraconazole (depicted in Fig. 3b).

Is the C-terminally S-tagged *nctA* allele (l. 231/2) that was used for interaction studies and ChIPseq functional?

Is the mutant deleted for the identified interactor Mot1 (l. 239) part of the comprehensive library?

The most relevant and interesting part of this manuscript relates to the NCT complex as global regulator of sterol biosynthesis – given the pleiotrophic phenotype displayed by corresponding mutants, can the authors speculate and discuss about its direct and indirect effects or modes of action?

In the section on ChIPseq, the numbers presented in the text somehow do not correlate to the ones in Fig. 4b, while the ones in Fig. 4a do not add up correctly.

For the sake of stringency, the part on elevated ergosterol levels in NCT complex mutants might be shifted to directly follow the speculation on increased flux through the biosynthetic pathway based on RNAseq data (l. 262), and the corresponding Fig. 5 could be incorporated into Fig. 3 as

panel e.

What was used as internal standard in the Western blot analysis of CDR1B levels (l. 335/6 and Fig. 6c)?

A brief comment on the fact that only the *nctA* null mutant (leaving out its *nctB* counterpart) was used in the immunogenic analyses (l. 343-358) should be provided.

Are there any references in the published literature supporting the "growing body of evidence that poor in vitro growth is not an absolute indicator of virulence defects." (l. 385)?

In the Discussion, citations for the screens using transcription factor null mutant libraries in various fungi (l. 394-396) are missing.

The rationale of increased Amphotericin B sensitivity based on upregulation of ergosterol biosynthesis (l. 432/3) is unclear.

The scenario of NctA/B-mediated azole resistance is highly relevant in a clinical context (l. 452-455) - has such an isolate been detected or described so far?

When speculating on the relevance for drug resistance in yeasts (l. 456-458), it is unclear which fungal organism is actually meant.

The Materials & Methods section appears as comprehensible and complete, only the use of oligonucleotides P1 and P4 in diagnostic PCR "to check the purity" (l. 486/7: of what?) might be explained.

Reviewer #2 (Remarks to the Author):

Furukawa et al. present a substantive study in which they aimed to identify unknown transcription factors whose rewiring could lead to resistance to azole drugs, the most important of the few drugs available for treating fungal disease with high mortality. They identify from a systematic screen, in the human fungal pathogen *Aspergillus fumigatus*, new transcription factors that regulate resistance and sensitivity to azole drugs. The authors identified 495 transcription factor genes in the genome sequence and were able to construct a transcription factor knockout library for 484 of these genes. They screened all 484 mutants for resistance and sensitivity to itraconazole, an important azole drug used for treatment of Aspergillosis and other fungal diseases. They identified 12 genes, including several that have not previously been associated with azole resistance or sensitivity, and assessed the effects of deletion of these genes on resistance and sensitivity to a range of key drugs used to treat fungal infections. They focus on the role of two novel genes *nctA* and *nctB*, which have the same deletion phenotype and encode transcription factors of the CBF/NFY class. The authors used RNA-seq to determine that the *nctA*Δ and *nctB*Δ mutants regulate essentially the same set of genes, used Co-IP-LCMS to demonstrate that the two proteins interact with each other and with an overlapping set of additional proteins including the TATA Binding Protein (TBP)-associated protein Mot1. They performed ChIP-seq studies to identify genome-wide direct targets for NctA. The authors then focused on the regulation by these transcription factors of the ergosterol biosynthesis pathway, which is the target of azole drugs. They demonstrated elevated ergosterol levels in the *nctA*Δ mutant. These levels, however, did not reflect the observed degree of resistance. They then identified an ABC transporter CDR1B that is a direct target of NctA demonstrated by ChIP-seq analysis, and show that both its mRNA and protein levels are elevated in the *nctA*Δ mutant.

This novel work provides a substantial and significant advance, both in the development of a near-complete transcription factor deletion library for dissection of regulatory mechanisms in filamentous fungi, and in its unmasking of new roles in azole sensitivity and resistance for several known transcription factors, as well as the identification of novel transcription factors including NctA and NctB, loss of which could lead to azole drug resistance in patients. The authors use multiple approaches – RNA-seq, ChIP-seq, co-immunoprecipitation, and specific assays – to robustly and convincingly reveal the underlying mechanism leading to azole resistance in the *nctA* and *nctB* mutants via dysregulation of the ergosterol pathway and an ABC transporter. The authors are careful to employ sufficient biological and technical replicates, and provide appropriate statistical support for their conclusions. The methods are presented clearly, with sufficient detail

that would allow the work to be repeated.

The identification of NctA and NctB, as well as the novel roles in azole resistance or susceptibility for other transcription factors will no doubt promote discussion and further work in this field. This work is of broad relevance to the spectrum of fungal diseases and therefore will be of broad interest to the medical mycology and fungal biology communities; it is also of interest to those in the transcriptional gene regulation field (and the availability of the transcription factor knockout library will provide an outstanding resource for further uncovering gene regulatory circuitry). The approach of screening a near-complete transcription factor deletion library is of general interest. In addition, azoles are also used as fungicides in agriculture, so I expect the readership to include those interested in plant health.

Suggested improvements

1. Lines 252-262: The authors highlight the regulation of the ergosterol biosynthesis pathway by NctA, and also note the role of decreased *erg7* in reducing diversion of 2,3-epoxysqualene from the ergosterol pathway. The authors do not mention that the siderophore biosynthesis genes (which divert mevalonate from the ergosterol pathway) are downregulated in the *nctAΔ* mutant and the genes that metabolize mevalonate *hmg2* and *erg12* are up-regulated, indicating a role in balancing of ergosterol and siderophore biosynthesis. This would also be worthy of comment.
2. The authors show interaction of both NctA and NctB via Co-IP LCMS with the TBP-associated transcription factor Mot1. The authors might comment that the *mot1* gene is bound by NctA in their ChIP-seq data and also is differentially expressed in *nctAΔ* in their RNA-seq data, indicating that regulation of Mot1 levels in addition to physical interaction with Mot1 forms part of the mechanism of NctA-NctB action.
3. It is indicated (lines 193 and 207) that the *nctAΔ* and *nctBΔ* mutants phenocopy each other for the drug resistance phenotypes, and this is convincing from the data presented. However, the growth phenotype of the *nctBΔ* is not shown or commented on – whether or not it shows a similar poor growth phenotype to *nctAΔ* (as shown in Fig. 2A) would be worthwhile stating, or potentially including an equivalent growth test image comparing WT, $\Delta nctA$ and $\Delta nctB$, possibly in supplementary figure 3, or elsewhere.

Minor edits

Overall the manuscript is very carefully prepared, particularly given the enormous amount of data. However, there are some minor edits needed:

Line 84: this sentence is clunky. It would read better as: "...repeat in the promoter, typically TR34 or TR46, with a secondary mutation L98H within its coding sequence commonly associated with TR34."

Line 122, 263, and title: "master regulator" is not a particularly good term to use, as this term has a specific meaning that is related to development regulation and usually describes positively acting transcription factors that are sufficient to direct cell fate and are at the top of a regulatory hierarchy (and therefore not regulated by other genes); see doi:10.4172/2157-7633.1000e114 . Although this term has, unfortunately, been used in other contexts by others, I suggest the authors consider an alternative description, such as a "key regulator" or "coordinator". The authors do show that NctA-NctB binds promoters of, and regulates expression of, other transcription factors that regulate drug resistance, which indicate the NC2 complex acts towards the top of the regulatory hierarchy, but NC2 is a repressor, which doesn't fit with the usual definition, and the regulation of *nctA* and *nctB* is not yet well established.

Line 150: "zinc finger" is ambiguous/unclear – do the authors mean Zn(II)₂Cys₆ (or C₆) zinc binuclear cluster (which are not usually referred to as zinc fingers), or does this include the various types of zinc finger (C₂H₂, GATA etc.)? The GATA zinc finger class, but not the C₂H₂ zinc finger class, are listed separately in Fig. S1b, so it is difficult for the reader to understand the use of "zinc finger". It would help clarify which Pfam family is referred to if the Pfam accession numbers for each domain class (e.g. PF00172 for Fungal Zn(2)-Cys(6) binuclear cluster domain) were indicated in Figure S1b.

Line 172: it is unclear what is meant by "higher level transcription factors" – additional explanation would help the reader.

Line 177-179: it would help the reader if the concentrations were presented in the sentence in order with respect to azole resistance and sensitivity – the order seems switched. "azole resistance and sensitivity... ..at itraconazole concentrations representing sub-minimum inhibitory concentrations... and above MIC...", yet sub-minimum corresponds to sensitivity and above MIC corresponds to resistance.

Line 383: "Fig. 8 d" should read "Fig. 8 e" (and the corresponding figure legend line 1066 (f) should be (e)).

Line 396: references for the functional screens of transcription factor deletion libraries should be included here.

Line 399: delete "in"

Line 403: "compared to yeast" would be more accurate as "compared to *S. cerevisiae*"

Line 407: "orthologues of ... three (ZipD, AreA and AtrR) are absent in yeast." This sentence could confuse the reader because for AreA there are functional orthologs, Gat1p and Gln3p. When using stringent cut-offs in reciprocal blast searches, AreA and Gat1p or Gln3p are not identified as strict sequence orthologues despite showing considerable sequence similarity, however, there is considerable evidence that Gat1p and Gln3p are the functional orthologues of AreA, and the domain architecture is conserved with Gat1p. The authors may consider rewording this sentence.

Line 414: "RNA pol II-dependent promoters"

Line 474: "Briefly, primers..."

Line 504: "as follows"

Line 587: "fumigatus"

Lines 710-899: the species names in the reference list should be in italics

Line 991: "it's" should be "its"

Fig. 6a: indicating that the values represent with a label in this panel "cdr1B expression" would make this more immediately clear to the reader.

Line 1066: (f) should be (e)

Fig. 9: The model presented in Fig. 9 is potentially helpful to the reader, but does not appear to be referred to in the text. Furthermore, the legend (Line1080) indicates that the NCT complex mediates transcriptional repression of the activator encoding *srbA* and *atrR* and activation of the negative regulator encoding *hapC*. The sharp and blunt arrows in the model appear inconsistent with the legend text (there is a sharp "activator" arrow to *atrR* but a blunt "repressor" arrow to *srbA* and *hapC*).

Line 1080 is unclear: this might be clearer as "the activator-encoding *srbA* and *atrR* genes, and activation of the negative regulator-encoding *hapC* gene"

Supplementary data 1, Tab1: It would be useful to the reader to include the genes identified in this study in the generic name list in this table (*atrR*, *zipD*, *adaB*, *gisB*, *rscE*, *nctA* and *nctB*).

Fig. S4 – it is unclear what is meant by "chDNA"; this abbreviation is not defined. (Ch is used as abbreviation in the legend for chromatin; gDNA or genomic DNA would be clearer); "nctA/nctB" would be clearer as "nctA or nctB".

Fig. S5 – panels c and d are not defined in the legend, so it is unclear what these represent. They are not referred to in the text, so may be dispensable(?).

Table S1 – it would help the reader if NctA and NctB were labeled as such in the table.

Supplementary data 2 Tab 5: "showin in gray" should be "shown in gray"

Supplementary data 3 Tab3: the tab label at the bottom is "No-drug" but this tab is the itraconazole treatment.

Reviewer #3 (Remarks to the Author):

In this manuscript, the authors report the generation of a transcription factor (TF) deletion library for the filamentous fungal pathogen, *Aspergillus fumigatus*. Subsequently, the authors utilize this novel tool to identify transcriptional regulators of triazole susceptibility. The recent expansion of

triazole resistance in *A. fumigatus* is concerning, as this class of antifungals is the mainstay of therapy. Although there is some conservation among yeast and filamentous fungal pathogens with respect to transcriptional controls, the work described herein succinctly lays out the need for study in the pathogen of interest. Therefore, the TF deletion library represents a first-of-its-kind advance for the field. The authors use this library to identify multiple TFs regulating triazole susceptibility and further characterize a conserved TF complex called the Negative Cofactor 2 complex. Data provided here show that loss of either component of the complex leads to triazole resistance as well as hypersusceptibility to cell wall stress. The authors utilize next-gen sequencing to identify ergosterol biosynthetic components and triazole exporters that may underlie the resistance phenotype noted. Data are also provided showing that loss of this TF does not affect pathogenic fitness. Overall, this is a very strong study and well-written manuscript. The authors report the development of an invaluable tool represented by the TF deletion library and have identified a novel regulator of triazole-induced stress responses. The statistical methods are appropriate and scientific rigor appears high. I have only a few points for the authors to consider:

1. The levels of gene expression changes for the *erg* genes and the additional transcriptional regulators, as well as ergosterol level changes, are all very modest. Additionally, the *cdr1B* transporter induction is well below levels reported to be associated with large MIC shifts. Therefore, it is hard to reconcile the high levels of resistance as being completely supported by these factors. The authors report, but do little to address, the exquisite sensitivity to cell wall stress in the NC2 mutants. Could there be a direct link between the cell wall composition and/or stress response in these mutants to azole resistance? For example, a recent study implicated cell wall dynamics in the cidal effect of triazoles on *A. fumigatus* (Nat Commun. 2018; 9: 3098.). Could alteration of these dynamics be at play? Are additional putative transporters upregulated upon loss of NC2? Alternative hypotheses addressing the high levels of azole resistance in the NC2 mutants need to be discussed.
2. Were the S-tagged NctA/B mutants functional? The authors should state this to further support the ChIPseq data, especially since the authors report low correlation between the RNAseq and ChIPseq data datasets.
3. As the authors state in the discussion, alteration of NctA/B activity may be possibly relevant to clinical resistance as fitness is unaffected. There are multiple publicly available genome datasets of clinically resistant and susceptible *A. fumigatus* isolates. Do the authors note mutations specific to resistant isolates in these genes or their promoters? Such findings would support this claim.

Response to reviewers:

We are grateful to all three reviewers for the critical comments and useful suggestions that have helped us to improve our paper considerably. As indicated in the responses that follow, we have taken all these comments and suggestions into account in the revised version of our paper. Please find below our point-by-point response to the reviewer's comments. Changes in the revised manuscript are indicated in blue.

Reviewer #1 (Remarks to the Author):

In this interesting and in part surprising manuscript, the authors describe their extensive efforts to unravel transcriptional regulation of relevant traits of the human pathogenic mould *Aspergillus fumigatus* on a global scale. Starting from the Hercules task of generating a comprehensive mutant strain library by targeting annotated transcription factor-encoding genes, the focus was set on means of resistance towards azole class antifungals. By this, it could be revealed that the Negative Cofactor 2 (NC2) orthologue of *A. fumigatus* might serve as master regulator of ergosterol biosynthesis and beyond: corresponding mutants display elevated immunogenicity but unaltered virulence in two murine infection models of invasive pulmonary aspergillosis. While the general aspect of describing a valuable resource for the first time is highly appreciated, the subsequent tasks are presented to some degree imprecisely to leave some questions that might be addressed (see comments below). Accordingly, several minor point of criticism exist that ought to be revised in a future version of this highly relevant piece of work that doubtlessly will be of high value not only for the fungal research community. In detail, specific issues are as follows.

Response to Reviewer 1

Comment 1

The Introduction would benefit from a concluding sentence summarizing the relevance of this very study.

Response

A short paragraph has been added as per the reviewer's request.

Comment 2

Why exactly does the lack of understanding azole resistance mechanisms prevent the development of combination therapies (l. 88-90)?

Response

We have modified this statement to clarify that this lack of understanding prevents direct targeting of resistance mechanisms.

Comment 3

In the Results, the strategy to identify TF-encoding genes is unclear: given that for 84 proteins no functional domain could be identified (l. 153/4), why were these included in the first place?

Response

We have modified this section for clarity and added a section in the Materials and Methods to clarify how the search for the TFs was conducted.

Comment 4

The gene targeting approach using NHEJ-deficient *A. fumigatus* recipients is incompletely referenced (l. 160).

Response

The references have been updated.

Comment 5

When selecting 12 mutants for genotype confirmation by Southern hybridization (l. 169), were any of the subsequently used strains among these? And were the genotypes of all the strains deleted for key regulators associated with azole resistance and sensitivity confirmed by this approach?

Response

The 12 mutants were chosen somewhat at random, however the *ΔnctA* isolate was included in this verification process. The details of the strains validated by Southern blot are given in Supplementary Data 1.

Comment 6

What is exactly meant by the term “higher level transcription factors” (l. 172/3) and why might the encoding genes be essential for *A. fumigatus*?

Response

This has been clarified in the text.

Comment 7

The fact that two master regulators of catabolite repression – CreA and AreA – appear to influence azole resistance (l. 184 and l. 189) raises the question on which type of culture medium and in the presence of which sources of carbon and nitrogen the corresponding assays were performed.

Response

This is an interesting question specifically as changes to methodologies and media composition can have a significant effect on the MIC of Itraconazole and many other drugs (Gomez-Lopez, et al., Journal of Clinical Microbiology. 2005, 43(3):1251-1255 and references therein). For this reason we chose to restrict our study to the use of a modified version of RPMI1640 which is employed in the clinical evaluation of azole resistant *A. fumigatus* isolates. This media includes glucose which *A. fumigatus* presumably uses as its primary carbon source and amino acids (including Glutamic acid) which are likely to be acting as primary nitrogen sources. The details of this well-defined culture media are presented in the materials and methods section however to aid clarity for the reader we have added a sentence to this effect in the main body of the text.

Comment 8

The general fitness (and its definition) of the key regulator deletants should be briefly described before elaborating on their phenotypes with respect to azoles or other drugs or cell wall stressors.

Response

Brief descriptions about the growth phenotypes of relevant null mutants have been added to the results section. Also graphs showing general growth fitness of the mutants and the pictures showing colonial growth of the mutants on *Aspergillus* complete medium (ACM) have been added as Supplementary Figure 3. Finally, the definition of the general fitness has been clarified in the materials and methods section.

Comment 9

What is actually the MIC for the recipient isolate MFIG001 that determines the sub-MIC and above MIC levels that were used for screening the transcription factor null library?

Response

The itraconazole MIC for the recipient strain is 0.5 mg/L. This information has been added to the results section.

Comment 10

When characterizing the CBF/NF-Y transcription factors NctA (is its paralogue AFUB_045980 also part of the knock-out library?) and NctB, why was the focus set on the $\Delta nctA$ strain, to what degrees were phenotypes displayed by the $\Delta nctA$ and $\Delta nctB$ mutants similar?

Response

Null mutants for *nctB* (AFUB_045980), and the *nctA* paralogue AFUB_058240 are in the knock-out library. The *nctB* null mutant phenocopies the *nctA* null mutant in all the basic screens we conducted, however we did not see any phenotype for the AFUB_058240 null. We have added an image that shows the gross phenotypic changes exhibited by the *nctB* null to Supplemental Figure 3, also we have added growth curves of the mutant in RPMI-1640 in Supplementary Figure 5. Our transcriptomic data also indicates that the roles of NctA and NctB are non-redundant with respect to each other. These facts led us to focus on the role of NctA in the more involved studies and more especially where we believed it was not ethically prudent to extend our study to the *nctB* null.

Comment 11

The description and comparison of the RNAseq datasets (l. 224-227) lacks the one in presence of itraconazole (depicted in Fig. 3b).

Response

Text to describe this data has been added to the results section.

Comment 12

Is the C-terminally S-tagged *nctA* allele (l. 231/2) that was used for interaction studies and ChIPseq functional?

Response

The strain incorporating the S-tagged *nctA* cassette, which replaces the native *nctA* gene is functional. This has been determined by comparing the growth of the isolate to the wild-type and null mutants and assessing the growth of the strain in the presence of itraconazole. In addition to these phenotypic characterizations, we have also confirmed functional expression of the S-tagged NctA protein using immunoprecipitation (IP) followed by Western blotting with a S-tag specific antibody. We have added a description for the phenotypes of the S-tagged NctA expressing strains in the results section and the data showing the phenotypes has been added to the Supplementary Figure 5.

Comment 13

Is the mutant deleted for the identified interactor Mot1 (l. 239) part of the comprehensive library?

Response

Mot1 (modifier of transcription 1 in *Saccharomyces cerevisiae*) was not identified in our original *in silico* identification of transcription factors as it does not have a DNA binding domain that has been annotated in the databases we have searched. Furthermore, it has been reported that Mot1 is a Swi2/Snf2-type ATPase and is not a DNA binding transcription factor (Adamkewicz et al., J Biol Chem., 2000, 14;275(28):21158-68). In *S. cerevisiae*, Mot1 functions to mediate a reaction that leads to dissociation of the TBP-DNA complex and hence should be considered as a modulator of transcription factor function rather than a transcription factor *per se*.

Comment 14

The most relevant and interesting part of this manuscript relates to the NCT complex as global regulator of sterol biosynthesis – given the pleiotropic phenotype displayed by corresponding mutants, can the authors speculate and discuss about its direct and indirect effects or modes of action?

Response

Text to describe the mechanistic basis of the NCT complex mediated transcriptional regulation of the ergosterol biosynthesis genes and the azole resistance has been added to the discussion section.

Comment 15

In the section on ChIPseq, the numbers presented in the text somehow do not correlate to the ones in Fig. 4b, while the ones in Fig. 4a do not add up correctly.

Response

We found that this discrepancy was originated from the fact that the different definitions were applied to select the “significant peaks” from the dataset when we made the figure. In the revised version, we have amended numbers and add a clear definition for the peaks used for this analysis in the figure legend of Figure 4. Also we have revised the text in this section, Figure 4b, Supplementary Figure 7 to correct these changes.

Comment 16

For the sake of stringency, the part on elevated ergosterol levels in NCT complex mutants might be shifted to directly follow the speculation on increased flux through the biosynthetic pathway based on RNAseq data (l. 262), and the corresponding Fig. 5 could be incorporated into Fig. 3 as panel e.

Response

We have considered this suggestion carefully however we feel as though the flow of the document maybe negatively affected by changing the order in which the data is presented.

Comment 17

What was used as internal standard in the Western blot analysis of CDR1B levels (l. 335/6 and Fig. 6c)?

Response

Total protein amount showing on the Ponceau S stained membrane (Figure 6) was used as the internal standard to quantify Cdr1B levels in the Western blotting analysis.

Comment 18

A brief comment on the fact that only the *nctA* null mutant (leaving out its $\Delta nctB$ counterpart) was used in the immunogenic analyses (l. 343-358) should be provided.

Response

We have added a sentence to state why we are focussing on *nctA*.

Comment 19

Are there any references in the published literature supporting the “growing body of evidence that poor *in vitro* growth is not an absolute indicator of virulence defects.” (l. 385)?

Response

At least 2 mutants of *A. fumigatus* have been shown to exhibit significant phenotypic defects *in vitro* whilst being seemingly unaffected *in vivo*. These include the null mutant of the mitogen acting protein kinase MpkA (Valiante et al, Fungal Genet Biol. 2008. 45(5):618-27) and a protein kinase C mutant with a Gly579Arg substitution (Rocha et al., PLoS ONE. 2015. 10(8):e0135195. In addition a previous study has demonstrated that although there is a correlation between growth rate and virulence, this correlation is not absolute (Paisley et al., Med Mycol. 2005. 43(5):397-401). These citations have been added to appropriate section of the text.

Comment 20

In the Discussion, citations for the screens using transcription factor null mutant libraries in various fungi (l. 394-396) are missing.

Response

These citations were unintentionally omitted and have been added in the revised manuscript.

Comment 21

The rationale of increased AmphoB sensitivity based on upregulation of ergosterol biosynthesis (l. 432/3) is unclear.

Response

We have attempted to clarify our statement and have added a further citation supporting this hypothesis.

Comment 22

The scenario of NctA/B-mediated azole resistance is highly relevant in a clinical context (l. 452-455) - has such an isolate been detected or described so far?

Response

To date, we are not aware that mutation of either *nctA* or *nctB* has been associated with azole resistance in clinical isolates of *A. fumigatus*. The genome sequences of a number of clinical and environmental isolates have been deposited in public databases (see FungiDB which enables rapid SNP evaluation of 48 sequenced strains; <https://fungidb.org>) and upon searching these data sets we have determined that 2 SNPs are present in *nctA* and 3 in *nctB*. Only one of these SNPs is non-synonymous SNP (NctA; I22V). Although 18 of the sequenced isolates that harbour the non-synonymous SNP are azole resistant, a further 5 are not. As relatively few sequenced isolates are in the public domain, a genetic association between drug resistance and SNPs in *nctA* and *nctB* is difficult to prove at this stage however we, in collaboration with others are extending our data sets and will be able to report further on this in the future. We have added a sentence to the discussion to highlight this.

Comment 23

When speculating on the relevance for drug resistance in yeasts (l. 456-458), it is unclear which fungal organism is actually meant.

Response

We have clarified our statement to highlight that we are citing literature from *C. albicans*.

Comment 24

The Materials & Methods section appears as comprehensible and complete, only the use of oligonucleotides P1 and P4 in diagnostic PCR “to check the purity” (l. 486/7: of what?) might be explained.

Response

We have added an explanation about the use of the oligonucleotide P1 and P4.

Reviewer #2 (Remarks to the Author):

Furukawa et al. present a substantive study in which they aimed to identify unknown transcription factors whose rewiring could lead to resistance to azole drugs, the most important of the few drugs available for treating fungal disease with high mortality. They identify from a systematic screen, in the human fungal pathogen *Aspergillus fumigatus*, new transcription factors that regulate resistance and sensitivity to azole drugs. The authors identified 495 transcription factor genes in the genome sequence and were able to construct a transcription factor knockout library for 484 of these genes. They screened all 484 mutants for resistance and sensitivity to itraconazole, an important azole drug used for treatment of Aspergillosis and other fungal diseases. They identified 12 genes, including several that have not previously been associated with azole resistance or sensitivity, and assessed the effects of deletion of these genes on resistance and sensitivity to a range of key drugs used to treat fungal infections. They focus on the role of two novel genes *nctA* and *nctB*, which have the same deletion phenotype and encode transcription factors of the CBF/NFY class. The authors used RNA-seq to determine that the $\Delta nctA$ and $\Delta nctB$ mutants regulate essentially the same set of genes, used Co-IP-LCMS to demonstrate that the two proteins interact with each other and with an overlapping set of additional proteins including the TATA Binding Protein (TBP)-associated protein Mot1. They performed ChIP-seq studies to identify genome-wide direct targets for NctA. The authors then focused on the regulation by these transcription factors of the ergosterol biosynthesis pathway, which is the target of azole drugs. They demonstrated elevated ergosterol levels in the $\Delta nctA$ mutant. These levels, however, did not reflect the observed degree of resistance. They then identified an ABC transporter CDR1B that is a direct target of NctA demonstrated by ChIP-seq analysis, and show that both its mRNA and protein levels are elevated in the $\Delta nctA$ mutant.

This novel work provides a substantial and significant advance, both in the development of a near-complete transcription factor deletion library for dissection of regulatory mechanisms in filamentous fungi, and in its unmasking of new roles in azole sensitivity and resistance for several known transcription factors, as well as the identification of novel transcription factors including NctA and NctB, loss of which could lead to azole drug resistance in patients. The authors use multiple approaches – RNA-seq, ChIP-seq, co-immunoprecipitation, and specific assays – to robustly and convincingly reveal the underlying mechanism leading to azole resistance in the *nctA* and *nctB* mutants via dysregulation of the ergosterol pathway and an ABC transporter. The authors are careful to employ sufficient biological and technical replicates, and provide appropriate statistical support for their conclusions. The methods are presented clearly, with sufficient detail that would allow the work to be repeated.

The identification of NctA and NctB, as well as the novel roles in azole resistance or susceptibility for other transcription factors will no doubt promote discussion and further work in this field. This work is of broad relevance to the spectrum of fungal diseases and therefore will be of broad interest to the medical mycology and fungal biology communities; it is also of interest to those in the transcriptional gene regulation field (and the availability of the transcription factor knockout library will provide an outstanding resource for further uncovering gene regulatory circuitry). The approach of screening a near-complete transcription factor deletion library is of general interest. In addition, azoles are also used as fungicides in agriculture, so I expect the readership to include those interested in plant health.

Response to Reviewer 2

Comment 1

Lines 252-262: The authors highlight the regulation of the ergosterol biosynthesis pathway by NctA, and also note the role of decreased *erg7* in reducing diversion of 2,3-epoxysqualene from the ergosterol pathway. The authors do not mention that the siderophore biosynthesis genes (which divert mevalonate from the ergosterol pathway) are downregulated in the *nctA mutant and the genes that*

metabolize mevalonate *hmg2* and *erg12* are up-regulated, indicating a role in balancing of ergosterol and siderophore biosynthesis. This would also be worthy of comment.

Response

The interaction between ergosterol biosynthesis and siderophore biosynthesis is clearly an interesting subject matter. Our need to balance brevity with a complete description of the study prevented us from discussing this extensively however given the prompt from the reviewer we have added a section to describe this link and its implication. We have also added gene expression profiles of siderophore biosynthesis genes in Supplemental Data 2.

Comment 2

The authors show interaction of both NctA and NctB via Co-IP LCMS with the TBP-associated transcription factor Mot1. The authors might comment that the *mot1* gene is bound by NctA in their ChIP-seq data and also is differentially expressed in *nctAΔ* in their RNA-seq data, indicating that regulation of Mot1 levels in addition to physical interaction with Mot1 forms part of the mechanism of NctA-NctB action.

Response

We have added a paragraph to describe the involvement of NCT complex in the regulation of *mot1* expression and its potential effects on modulating the regulatory function of NCT complex in azole resistance. We have also added the expression levels of *mot1* in the *nctA* and the *nctB* null mutants in Supplementary Data 2. .

Comment 3

It is indicated (lines 193 and 207) that the *nctAΔ* and *nctBΔ* mutants phenocopy each other for the drug resistance phenotypes, and this is convincing from the data presented. However, the growth phenotype of the *nctBΔ* is not shown or commented on – whether or not it shows a similar poor growth phenotype to *nctAΔ* (as shown in Fig. 2A) would be worthwhile stating, or potentially including an equivalent growth test image comparing WT, $\Delta nctA$ and $\Delta nctB$, possibly in supplementary figure 3, or elsewhere.

Response

We agree that this omission should be rectified. A figure showing colonial growth of the $\Delta nctB$ mutant has been included in Supplementary Figure 3 together with the other azole resistance associated regulators. We have also added a time-course growth profiling of the $\Delta nctA$ and the $\Delta nctB$ null mutant in RPMI-1640 culture medium in Supplemental Figure 5 to further support this statement.

Comment 4

Line 122, 263, and title: “master regulator” is not a particularly good term to use, as this term has a specific meaning that is related to development regulation and usually describes positively acting transcription factors that are sufficient to direct cell fate and are at the top of a regulatory hierarchy (and therefore not regulated by other genes); see doi:10.4172/2157-7633.1000e114 . Although this term has, unfortunately, been used in other contexts by others, I suggest the authors consider an alternative description, such as a “key regulator” or “coordinator”. The authors do show that NctA-NctB binds promoters of, and regulates expression of, other transcription factors that regulate drug resistance, which indicate the NC2 complex acts towards the top of the regulatory hierarchy, but NC2

is a repressor, which doesn't fit with the usual definition, and the regulation of nctA and nctB is not yet well established.

Response

We have modified our phraseology and now use the term “key regulator” as suggested.

Comment 5

Line 150: “zinc finger” is ambiguous/unclear – do the authors mean Zn(II)₂Cys₆ (or C₆) zinc binuclear cluster (which are not usually referred to as zinc fingers), or does this include the various types of zinc finger (C₂H₂, GATA etc.)? The GATA zinc finger class, but not the C₂H₂ zinc finger class, are listed separately in Fig. S1b, so it is difficult for the reader to understand the use of “zinc finger”. It would help clarify which Pfam family is referred to if the Pfam accession numbers for each domain class (e.g. PF00172 for Fungal Zn(2)-Cys(6) binuclear cluster domain) were indicated in Figure S1b.

Response

We have indicated the corresponding Pfam accession ID to each domain class in the revised Supplementary Figure 1b.

Comment 6

Line 172: it is unclear what is meant by “higher level transcription factors” – additional explanation would help the reader.

Response

An additional explanation to higher level transcription factors has been added to the revised manuscript.

Comment 7

Line 177-179: it would help the reader if the concentrations were presented in the sentence in order with respect to azole resistance and sensitivity – the order seems switched. “azole resistance and sensitivity... ..at itraconazole concentrations representing sub-minimum inhibitory concentrations... and above MIC...”, yet sub-minimum corresponds to sensitivity and above MIC corresponds to resistance.

Response

The text has been revised to corresponding to the order of azole resistance and sensitivity.

Comments 8

Line 407: “orthologues of ... three (ZipD, AreA and AtrR) are absent in yeast.” This sentence could confuse the reader because for AreA there are functional orthologs, Gat1p and Gln3p. When using stringent cut-offs in reciprocal blast searches, AreA and Gat1p or Gln3p are not identified as strict sequence orthologues despite showing considerable sequence similarity, however, there is considerable evidence that Gat1p and Gln3p are the functional orthologues of AreA, and the domain architecture is conserved with Gat1p. The authors may consider rewording this sentence.

Response

We have revised this sentence and included *S. cerevisiae* Gat1p and Gln3P as a functional orthologues of AreA.

Comment 9

Fig. 9: The model presented in Fig. 9 is potentially helpful to the reader, but does not appear to be referred to in the text. Furthermore, the legend (Line1080) indicates that the NCT complex mediates transcriptional repression of the activator encoding *srbA* and *atrR* and activation of the negative regulator encoding *hapC*. The sharp and blunt arrows in the model appear inconsistent with the legend text (there is a sharp “activator” arrow to *atrR* but a blunt “repressor” arrow to *srbA* and *hapC*).

Response

We have cited Figure 9 in the main text of the revised manuscript. Also, we have revised Figure 9 according to the suggestions.

Comment 10

Line 403: “compared to yeast” would be more accurate as “compared to *S. cerevisiae*”

Response

“yeast” has been revised to “yeasts” since our comparison includes not only *S. cerevisiae* but also the other pathogenic yeasts.

Comment11

Fig. 6a: indicating that the values represent with a label in this panel “*cdr1B* expression” would make this more immediately clear to the reader.

Response

We have revised Figure 6a according to the suggestions and the figure preparation guidelines of Nature communications (Figures do not allow to contain tables therefore the table in Fig6a has been replaced with a bar graph).

Comment 12

Minor Comments: Overall the manuscript is very carefully prepared, particularly given the enormous amount of data. However, there are some minor edits needed:

Line 84: this sentence is clunky. It would read better as: “...repeat in the promoter, typically TR34 or TR46, with a secondary mutation L98H within its coding sequence commonly associated with TR34.”

Line 383: “Fig. 8 d” should read “Fig. 8 e” (and the corresponding figure legend line 1066 (f) should be (e)).

Line 396: references for the functional screens of transcription factor deletion libraries should be included here.

Line 399: delete “in”

Line 414: “RNA pol II-dependent promoters”

Line 474: “Briefly, primers...”

Line 504: “as follows”

Line 587: “fumigatus”

Lines 710-899: the species names in the reference list should be in italics

Line 991: “it’s” should be “its”

Line 1066: (f) should be (e).

Line 1080 is unclear: this might be clearer as “the activator-encoding *srbA* and *atrR* genes, and activation of the negative regulator-encoding *hapC* gene”

Supplementary data 1, Tab1: It would be useful to the reader to include the genes identified in this study in the generic name list in this table (*atrR*, *zipD*, *adaB*, *gisB*, *rscE*, *nctA* and *nctB*).

Fig. S4 – it is unclear what is meant by “chDNA”; this abbreviation is not defined. (Ch is used as abbreviation in the legend for chromatin; gDNA or genomic DNA would be clearer); “nctA/nctB” would be clearer as “nctA or nctB”.

Fig. S5 – panels c and d are not defined in the legend, so it is unclear what these represent. They are not referred to in the text, so may be dispensable(?).

Table S1 – it would help the reader if NctA and NctB were labeled as such in the table.

Supplementary data 2 Tab 5: “showin in gray” should be “shown in gray”

Supplementary data 3 Tab3: the tab label at the bottom is “No-drug” but this tab is the itraconazole treatment.

Response

We have addressed and made changes for all of the suggested points in the revised version of the manuscript.

Reviewer #3 (Remarks to the Author):

In this manuscript, the authors report the generation of a transcription factor (TF) deletion library for the filamentous fungal pathogen, *Aspergillus fumigatus*. Subsequently, the authors utilize this novel tool to identify transcriptional regulators of triazole susceptibility. The recent expansion of triazole resistance in *A. fumigatus* is concerning, as this class of antifungals is the mainstay of therapy. Although there is some conservation among yeast and filamentous fungal pathogens with respect to transcriptional controls, the work described herein succinctly lays out the need for study in the pathogen of interest. Therefore, the TF deletion library represents a first-of-its-kind advance for the field. The authors use this library to identify multiple TFs regulating triazole susceptibility and further characterize a conserved TF complex called the Negative Cofactor 2 complex. Data provided here show that loss of either component of the complex leads to triazole resistance as well as hypersusceptibility to cell wall stress. The authors utilize next-gen sequencing to identify ergosterol biosynthetic components and triazole exporters that may underlie the resistance phenotype noted. Data are also provided showing that loss of this TF does not affect pathogenic fitness. Overall, this is a very strong study and well-written manuscript. The authors report the development of an invaluable tool represented by the TF deletion library and have identified a novel regulator of triazole-induced stress responses. The statistical methods are appropriate and scientific rigor appears high. I have only a few points for the authors to consider:

Response to Reviewer 3

Comment 1

The levels of gene expression changes for the *erg* genes and the additional transcriptional regulators, as well as ergosterol level changes, are all very modest. Additionally, the *cdr1B* transporter induction is well below levels reported to be associated with large MIC shifts. Therefore, it is hard to reconcile the high levels of resistance as being completely supported by these factors. The authors report, but do little to address, the exquisite sensitivity to cell wall stress in the NC2 mutants. Could there be a direct link between the cell wall composition and/or stress response in these mutants to azole resistance? For example, a recent study implicated cell wall dynamics in the cidal effect of triazoles on *A. fumigatus* (Nat Commun. 2018; 9: 3098.). Could alteration of these dynamics be at play? Are additional putative transporters upregulated upon loss of NC2? Alternative hypotheses addressing the high levels of azole resistance in the NC2 mutants need to be discussed.

Response

Many thanks to the reviewer for this prompt. We have added the following paragraph to the text:

“We have shown that loss of the NC2 complex leads to a large increase in MIC to the azoles, most notably for posaconazole where we observed a shift in MIC from 0.25 mg/L to >16 mg/L. This phenotype is associated with transcriptional dysregulation of the ergosterol biosynthetic pathway, an increase in cellular ergosterol levels (c. 1.6 fold) and an increase in levels of the Cdr1B azole transporter (2.4 fold). Although the increase in sterol levels is rather modest and apparently out of keeping with the large increase in MIC, recent evidence does not support a direct linear relationship between ergosterol levels and azole tolerance. Ryback *et al*, described mutation in the sterol sensing domain of *hmg1* (*hmg1*^{F262del}) that results in an apparent de-repression of ergosterol biosynthesis leading to a modest increase (1.6 fold) in ergosterol levels but a much higher relative increase in MIC to itraconazole and isavuconazole (8 fold) (mBio 10:e00437-19). The reason for this incongruity is unclear however studies in *S. cerevisiae* have highlighted that changes in the composition of the plasma membrane affect the function of transporters. For example, depletion in sterol levels caused by loss of *erg4* or *erg6*, leads to a reduction in the activity of the multi-drug resistance transporter Pdr5 (Kodedova and Sychrova, PLoS ONE, 2015, 10(9): e0139306) and ergosterol is required to correctly localise the azole exporter Cdr1p in *C. albicans* (Pasrija *et al.*, Antimicrob Agents Chemother. 2008. 52(2):694-704). This leads us to speculate that even relatively small increases in ergosterol content in

the cell membrane may lead to large increases in azole resistance via an indirect effect on azole transporter levels. Given the pleiotropic nature of the NC2 complex we cannot exclude that further factors may be contributing to the high levels of azole resistance evident in the *nctA* and *nctB* null mutants especially in light of our evidence showing that they are hypersensitive to the cell wall acting agents and recent data showing a link between reductions in β -1,3-glucan synthesis and the delayed fungicidal effects of voriconazole (Geißel et al., Nat Commun. 2018. 9(1):3098).”

Comment 2

Were the S-tagged NctA/B mutants functional? The authors should state this to further support the ChIPseq data, especially since the authors report low correlation between the RNAseq and ChIPseq data datasets.

Response

This is covered in the answer to the question posed by reviewer 1.

Comment3

As the authors state in the discussion, alteration of NctA/B activity may be possibly relevant to clinical resistance as fitness is unaffected. There are multiple publicly available genome datasets of clinically resistant and susceptible *A. fumigatus* isolates. Do the authors note mutations specific to resistant isolates in these genes or their promoters? Such findings would support this claim.

Response

This is covered in the answer to the question posed by reviewer 1.

REVIEWERS' COMMENTS:

Reviewer #1 (Remarks to the Author):

[No further comments for author.]